# Dealing with Synthetic Data Contamination in Online Continual Learning

**Maorong Wang**[1]   **Nicolas Michel**[1,2]   **Jiafeng Mao**[1]   **Toshihiko Yamasaki**[1]

[1]The University of Tokyo   [2]Univ Gustave Eiffel, CNRS, LIGM

`{ma_wang, yamasaki}@cvm.t.u-tokyo.ac.jp`
`nicolas.michel@univ-eiffel.fr`
`mao@hal.t.u-tokyo.ac.jp`

## Abstract

Image generation has shown remarkable results in generating high-fidelity realistic images, in particular with the advancement of diffusion-based models. However, the prevalence of AI-generated images may have side effects for the machine learning community that are not clearly identified. Meanwhile, the success of deep learning in computer vision is driven by the massive dataset collected on the Internet. The extensive quantity of synthetic data being added to the Internet would become an obstacle for future researchers to collect "clean" datasets without AI-generated content. Prior research has shown that using datasets contaminated by synthetic images may result in performance degradation when used for training. In this paper, we investigate the potential impact of contaminated datasets on Online Continual Learning (CL) research. We experimentally show that contaminated datasets might hinder the training of existing online CL methods. Also, we propose **E**ntropy **S**election with **R**eal-synthetic similarity **M**aximization (ESRM), a method to alleviate the performance deterioration caused by synthetic images when training online CL models. Experiments show that our method can significantly alleviate performance deterioration, especially when the contamination is severe. For reproducibility, the source code of our work is available at `https://github.com/maorong-wang/ESRM`.

## 1 Introduction

Continual Learning (CL) [10, 31, 41, 12] solves the problem of learning from a sequence of ever-emerging machine-learning tasks without forgetting previously learned knowledge. Defined by learning manners, CL can be classified into two categories [4]: *offline* CL and *online* CL. In offline CL (*i.e.* conventional CL), the learners can access the training dataset on **current** task multiple times before proceeding to the next task. In online CL, the training data also comes in a continual data stream, and the continual learner only sees the training data once. Besides learning manners, there are also three typical CL settings [38]: Task-Incremental Learning (TIL), Domain-Incremental Learning (DIL), and Class-Incremental Learning (CIL). In this paper, we investigate the more challenging CIL setting in the online CL manner.

Image generation with deep generative models has shown remarkable success. Thanks to denoising diffusion models [19, 34], Internet users are capable of generating high-fidelity and realistic images within several seconds. Despite the astonishing quality of those images to human eyes, research has shown that AI-generated content may be harmful when used to train machine learning models, leading to potential performance deterioration [18, 28], bias amplification [8], loss of diversity [28], etc.

38th Conference on Neural Information Processing Systems (NeurIPS 2024).

Recently, it has become a trend for researchers to collect datasets from the Internet, and synthetic data contamination would become a potential threat to the CL community. Moreover, the online CL is particularly affected as assessing the soundness of data in the online scenario is impractical. In this work, we first aim to investigate how this new form of dataset contamination might affect the existing Online CL methods. Then, we empirically observe the characteristics of synthetic data when used to train online CL models and form four observations of synthetic data properties in online CL, which might be of interest to the community. Moreover, we investigate synthetic data properties and exhibit specific differences in terms of entropy and representations when compared to real data. Guided by these properties, we propose **E**ntropy **S**election with **R**eal-synthetic similarity **M**aximization (ESRM), a method to alleviate the performance degradation caused by the synthetic contamination. As a replay-based method, ESRM consists of two key components: Entropy Selection (ES) and Real-synthetic similarity Maximization (RM). ES selects more realistic samples in the memory buffer to alleviate catastrophic forgetting. RM is a contrastive learning based optimization strategy, that aims to alleviate the performance artifact caused by synthetic data contamination.

The major contribution of this paper can be summarized as follows:

- We investigate the potential impact of synthetic data contamination on existing online CL methods and outline four observations regarding the properties of synthetic data in continual scenarios.
- We propose ESRM, a method to alleviate the performance deterioration caused by synthetic data contamination.
- Comprehensive experiments show that ESRM can successfully mitigate the performance deterioration caused by synthetic contamination, especially when the contamination is severe.

## 2 Related Work

**Synthetic data contamination.** Recently, diffusion models [19, 34] have achieved high-fidelity image generation and surpassed GANs in terms of image quality and diversity. Likewise, text-to-image generation based on diffusion models can generate astonishing images that faithfully follow the users' text instructions. Furthermore, these generative models demonstrate excellent extrapolation capabilities (*i.e.*, meaningfully combining concepts that would be nearly impossible to combine in reality), such as "a photo of an astronaut riding a house". Various generative models are open-sourced to the public, and users can use these models to generate realistic images in seconds. However, while people are appreciating the new format of art and flooding the Internet with fabulous images, such synthetic images are difficult to differentiate from the real ones. Therefore, these generated images are becoming a potential source of contamination for the future datasets collected from the Internet. Research has proven that synthetic data contamination may lead to a significant performance drop when supervising machine learning models in non-continual scenarios [18, 28]. Also, training deep models with such a contaminated dataset may give rise to bias amplification [8] and loss of diversity [28]. To tackle the issue caused by the synthetic contamination, researchers have proposed different strategies to detect the synthetic data with deep learning based detectors [30, 27]. However, the challenge brought by synthetic data contamination to the CL community is exclusive, and it is even more problematic in the online scenario since it is almost impossible to assess the quality of the training data, due to the unique challenge brought by the online setting.

**Continual Learning.** The mainstream CL strategies can be classified into four categories: regularization-based, parameter-isolation-based, prompt-based, and replay-based. Regularization-based methods [7, 26, 1, 23, 46] design and apply extra regularization terms to balance the learning and forgetting of CL learners. Parameter-isolation-based methods [14, 36, 37, 35, 3] tackle the CL problem by allocating task-specific parameters. Prompt-based methods [44, 43] take the idea of prompt learning and use prompt pools against catastrophic forgetting. Replay-based methods [33, 5, 6, 16, 17, 45] store a small portion of historical data with a memory buffer. Among all of the strategies, replay-based methods have prevailed in Online CL with better performance and simplicity. Early work [33] proposed **Experience Replay (ER)**, suggesting using a random replay buffer to alleviate catastrophic forgetting. **Dark Experience Replay (DER++)** [5] proposes to store the logits in the memory buffer and leverage the stored logits as dark knowledge to extend ER. **ER-ACE** [6] is a variant of ER with asymmetric cross-entropy loss. **OCM** [16] alleviates catastrophic

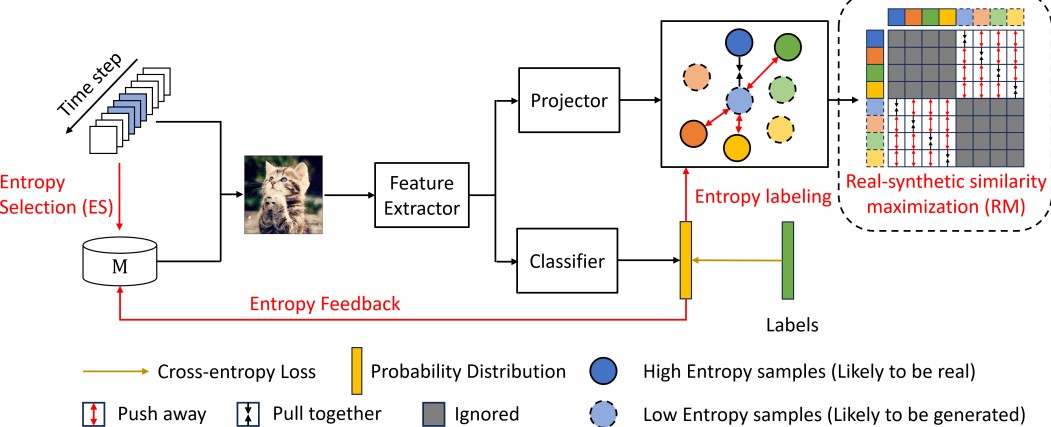

Figure 1: Overview of proposed ESRM framework for online CL. The proposed ESRM framework has two main components: Entropy Selection (ES) and Real-synthetic similarity Maximization (RM). Motivated by Obs. 2 and Obs. 3, ES is a buffer management strategy designed to use entropy as a criterion to select more real samples in the memory buffer, thereby alleviating catastrophic forgetting and performance degradation caused by the contamination. RM aims to bridge the embedding gap between synthetic and real data, as noted in Obs. 4, using a contrastive learning technique.

forgetting by maximizing the mutual information between current and past representations. **GSA** [17] addresses the cross-task class discrimination problem with gradient self-adaption. **OnPro** [45] solves the shortcut learning problem with online prototype learning. In this paper, these methods are used as baselines to assess the impact of synthetic data contamination.

## 3 Preliminary

### 3.1 Synthetic dataset generation

To simulate a dataset contaminated with synthetic data, we employed five diffusion-based models to generate synthetic counterparts of the original datasets. These twin datasets contain the same number of images and the same classes, while all images are synthetic. The models used in generation include Stable Diffusion XL [34], Stable Diffusion v1.4, Stable Diffusion v2.1, VQDM [15], and GLIDE [29]. The synthetic data contamination was simulated across four benchmark datasets used in online CL, including CIFAR-10 [24], CIFAR-100 [24], TinyImageNet [25], and ImageNet-100 [13, 20].

The generation of synthetic twin datasets is guided by the category names of the original dataset. For each class, we devised a simple yet effective prompt. For instance, for the class "helicopter", we employed the prompt "an image of a helicopter". After the generation by the diffusion model, we adjusted the image size to match that of the original dataset.

In the experiments, we design two different settings: **a)** all the synthetic twin datasets are generated from Stable Diffusion XL, one of the state-of-the-art generative models; and **b)** Synthetic images are generated with the aforementioned five diffusion models, with each model contributing 20% to the synthetic dataset.

For setting **a)**, we denote the generated dataset as SDXL-C10, SDXL-C100, SDXL-Tiny, and SDXL-In100, respectively, while for setting **b)**, we denote the generated dataset as Mix-C10, Mix-C100, and Mix-Tiny. Some examples of generated images and more detailed information about the synthetic twin dataset generation can be found in Appendix D.2. Notably, the generation results in Fig. 10 reveal the lack of diversity for synthetic data compared with the real data.

### 3.2 Simulation of synthetic data contamination

To simulate the contamination by the synthetic data, we substitute a portion $P$ of the original dataset with its synthetic twin, where $P$ is the contamination ratio. We designate these contaminated datasets using specific notations. For example, we denote the CIFAR-100 contaminated by SDXL-C100 as C100/SDXL. More details about the simulation of contamination are included in Appendix D.3. In

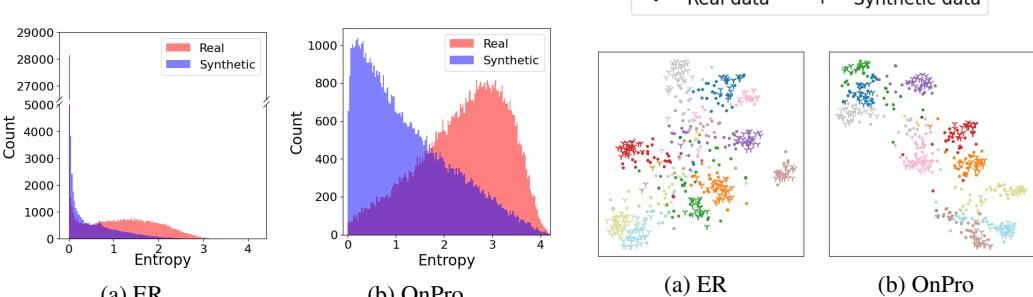

Figure 2: The entropy distribution of the training dataset produced by ER and OnPro on In-100/SDXL ($P = 50\%$) at the end of the training.

Figure 3: T-SNE visualization of the memory data at the end of training on In-100/SDXL ($P = 50\%$). For clarity, only the first 10 classes are visualized.

the following sections, we will investigate the effect of synthetic contamination with these simulated datasets.

## 4 Synthetic Data Contamination in Online CL

In this section, we explore the potential impact of synthetic data contamination on the existing online CL methods and exhibit specific properties of synthetic data.

### 4.1 Contamination as a cause of performance degradation

When dealing with synthetic data contamination, it is crucial to measure the impact of such contamination on existing methods. In that sense, we train ResNet-18 models in the online continual setting on the contaminated dataset C10/SDXL, C100/SDXL, Tiny/SDXL, and In-100/SDXL with representative online CL methods. We observed that as the contamination ratio $P$ increases, the performance of all existing methods drops significantly, as shown in Table 2. Detailed information about the experiment settings can be found in Sec. 6. With such experiments, we can form the following observation:

**Observation 1** *As a source of potential contamination, synthetic data is harmful to the performance of existing online CL methods. Performance degradation increases as contamination becomes more severe.*

### 4.2 Detecting synthetic data matters

Replay-based methods are characterized by the existence of a replay buffer, which helps alleviate forgetting and implicitly improves network plasticity [42]. The quality of data in the replay buffer intuitively influences network performance. Motivated by Obs. 1, we hypothesize that the presence of synthetic data in the replay buffer might degrade performance. We trained ER on the C100/SDXL dataset, with extra information on the samples' synthetic status (*i.e.*, whether the image is real or synthetic). We em-

| Method | Memory strategy | $P = 70\%$ Acc. ↑ | $P = 80\%$ Acc. ↑ |
|--------|-----------------|-----------------|-----------------|
| ER | Real Only | $38.44_{\pm0.90}$ | $38.13_{\pm1.34}$ |
| | Random | $32.82_{\pm1.62}$ | $31.33_{\pm1.33}$ |
| | Synthetic Only | $22.45_{\pm1.88}$ | $22.01_{\pm1.35}$ |

Table 1: The performance of ER with different memory strategies on C100/SDXL dataset, with different contamination ratio $P$.

ployed two memory strategies: storing only real data in the replay buffer and storing only synthetic data. The results showed in Table 1 indicate that knowing the synthetic status and storing only real data in the replay buffer can achieve performance on par with the no-contamination scenario, even at high contamination ratios. For instance, at a contamination ratio of $P = 80\%$, ER achieves an accuracy of 38.13% on C100/SDXL when only real images are stored in the replay buffer. This result is comparable to the accuracy achieved when training on the clean CIFAR-100 dataset (38.70%), which contains four times more real data. These findings also demonstrate the potential of synthetic models to enhance performance through data generation and augmentation in an online continual learning scenario.

> **Observation 2** *The memory buffer plays a key role in replay-based methods, and storing real samples in the memory buffer is effective against performance degradation caused by contamination.*

### 4.3 Synthetic data properties

**Lower entropy distribution.** One noteworthy characteristic of synthetic data is its lower entropy distribution compared to real data, as observed from the perspective of continual learners. Fig. 2 shows the entropy distribution produced by a representative method ER and a state-of-the-art method OnPro when trained on the contaminated dataset. The values in the histogram are calculated at the end of the training on the whole training dataset (In-100/SDXL, $P = 50\%$). From the figure, we can spot a salient distribution difference between synthetic data and real data. Moreover, the synthetic samples have a peak in entropy distribution close to 0. Extra entropy histogram with other baselines can be found in Appendix C.1. Thus, we conclude with another finding:

> **Observation 3** *Compared with real data, synthetic data entropy tends to be lower.*

**Feature gap in the embedding space.** We find another intriguing property of synthetic data in the feature embeddings. Fig. 3 shows the t-SNE [39] visualization of the memory data produced by ER and OnPro on the In-100/SDXL dataset. For clarity, we only visualize the embeddings of the first 10 classes. With the synthetic data contamination, the features of the synthetic samples are better clustered than the real data. The pattern of the clustering of synthetic data indicates the ease of classification, which is on par with the limited diversity of synthetic data (cf. Fig. 10), and the low entropy distribution (cf. Obs. 3). Moreover, the embeddings of real data are inferior and fail to align with the superior embeddings of synthetic samples, which explains the performance degradation of inference on real test datasets. Extra visualization produced by other baselines is illustrated in Appendix C.2.

> **Observation 4** *With the limited diversity of synthetic data, the synthetic data are better clustered than the real data, leading to a misalignment in the embedding space between synthetic samples and real samples. Such misalignment likely contributes to performance deterioration.*

## 5 Proposed Method

Fig. 1 presents the main framework of our proposed ESRM to alleviate the performance degradation caused by synthetic data contamination. In this section, we introduce the two components of ESRM: Entropy Selection (ES) and Real-synthetic similarity Maximization (RM). Then, we explain the whole ESRM framework.

### 5.1 Entropy selection

The introduced ES aims to select more representative samples in the memory buffer. For replay-based methods, having high-quality samples in the memory buffer helps alleviate forgetting and achieve better overall performance. As per Obs. 2, selecting real images into the memory buffer can provide more representative and reliable features and therefore lead to better performance. Motivated by this, we propose ES, a memory management strategy. Guided by Obs. 3 (synthetic data has lower entropy distributions), the core idea of ES is to select more real samples in the memory buffer based on the entropy distribution of the current batch.

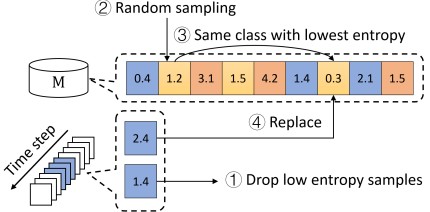

Figure 4: Overview of the proposed Entropy Selection strategy. The color of the samples indicates the class, and the number in the samples represents the entropy predicted by the learner.

At the beginning of the training, ES initializes an empty buffer. When a new batch comes, ES drops 50% of the batch samples with lower entropy, and stores the remaining samples, along with their entropy of the prediction. Once the buffer is full, as shown in Fig. 4, ES takes four steps to replace the elements in the buffer. Firstly, ES drops 50% of low entropy samples in the incoming batch. Then, ES uses Reservoir Sampling [21, 40] to decide whether to keep or discard the remaining incoming samples. If Reservoir Sampling decides to keep the incoming sample, it will nominate a buffer sample. After that, ES checks the class of the nominated sample and chooses a sample of the same class with the lowest

entropy in the memory buffer. And finally, ES replaces the chosen memory sample with the incoming sample, along with its entropy. Moreover, all entropy values in the memory buffer are updated by the current model's prediction at the end of each task. The pseudo-code of ES is given in Appendix B.

## 5.2 Real-synthetic similarity maximization

Prior to introducing the loss function of RM, we present the network structures of our ESRM. As shown in Fig. 1, the continual learner consists of three components: a feature extractor $f$, a projection head $g$, and a classifier $\phi$. The output dimension of the projection head $g$ is set to 128. For each sample $x$ from incoming data stream $X^{new}$, the projected embedding $z$ can be formulated as:

$$z = g(f(x)).\tag{1}$$

As per Obs. 4, the gap in the embedding space might be a cause for the performance degradation in the contamination setting. To tackle this issue, we propose RM. The main idea of RM is to maximize the cosine similarity between the features of real and synthetic data. Motivated by the seminal supervised contrastive loss [22], we propose our variant RM loss.

RM aims to maximize the similarity between two groups of data $X_1$ and $X_2$. We define the loss to match the similarity of samples in group $X_1$ to group $X_2$ as:

$$\mathcal{L}_M(X_1, X_2) = \sum_{i \in I_1} \frac{-1}{|P(i)|} \sum_{p \in P(i)} log \frac{exp(z_i \cdot z_p/\tau)}{\sum_{d \in I_2} exp(z_i \cdot z_d/\tau)},\tag{2}$$

where $I_1 = \{i : x_i \in X_1\}, I_2 = \{d : x_d \in X_2\}$ are the set of the indices of $X_1, X_2$, respectively. And $P(i) = \{p \in X_2 : y_p = y_i\}$ is the set of the indices of positive samples in group $X_2$, which share the same class with $x_i$. $\tau$ is the temperature hyperparameter which is set to 0.07. Since $\mathcal{L}_M(X_1, X_2)$ only optimize samples in the group $X_1$, in the optimization, it is used together with $\mathcal{L}_M(X_2, X_1)$. Due to RM aims to maximize the inter-group similarity between the $X_1$ group and the $X_2$ group, we do not need to perform augmentations as shown in similar work, such as SimCLR [9] and SupCon [22]. Intuitively, the way to handle the similarity matrix is illustrated in Fig.1.

For an incoming batch $X^{new}$, we use entropy criteria to split it into two groups $X^{new}_+$ and $X^{new}_-$ of the same size. Group $X^{new}_+$ includes 50% of the samples with the highest entropy, and as per Obs. 3, tend to contain more real images. On the contrary, group $X^{new}_-$ contains low entropy samples which tend to be synthetic. To alleviate the gap in feature embeddings mentioned in Obs. 4, we maximize the inter-group similarity between $X^{new}_+$ and $X^{new}_-$ with $\mathcal{L}_M(X^{new}_+, X^{new}_-)$ and $\mathcal{L}_M(X^{new}_-, X^{new}_+)$. Moreover, to align the holistic feature embeddings between the stream data $X^{new}$ and memory data $X^{mem}$, we also applied $\mathcal{L}_M(X^{new}, X^{mem})$ and $\mathcal{L}_M(X^{mem}, X^{new})$ in the loss function.

Thus, the proposed RM to alleviate the feature gap in Obs. 4 can be achieved by employing $\mathcal{L}_{RM}$:

$$\mathcal{L}_{RM} = \mathcal{L}_M(X^{new}_+, X^{new}_-) + \mathcal{L}_M(X^{new}_-, X^{new}_+) + \mathcal{L}_M(X^{new}, X^{mem}) + \mathcal{L}_M(X^{mem}, X^{new}).\tag{3}$$

## 5.3 Overall framework of ESRM

The overall framework of ESRM is shown in Fig. 1. Besides ES and RM, following [42], ESRM employs a self-distillation technique to alleviate the overconfidence problem of the replay-based methods. For the combined batch $X = (X^{new}, X^{mem})$, we apply the self-distillation as:

$$\mathcal{L}_{SDC} = D_{KL}(\phi(f(X))/t, \overline{\phi(f(aug(X)))}/t),\tag{4}$$

where $D_{KL}(\cdot)$ is the Kullback-Leibler divergence, $t$ is another temperature hyperparameter which is set to 4, $\overline{\phi(f(aug(X)))}$ is the fixed copy of $\phi(f(aug(X)))$, without gradient propagation, and $aug(\cdot)$ is the data augmentation used in the training process, with detailed information in Appendix D.5.

Thus, the total loss of ESRM can be formulated as:

$$\mathcal{L}_{ESRM} = \mathcal{L}_{CE} + \lambda_1 \mathcal{L}_{SDC} + \lambda_2 \mathcal{L}_{RM},\tag{5}$$

where $\mathcal{L}_{CE} = CE(\phi(f(X)), y) + CE(\phi(f(aug(X))), y)$ is the cross-entropy loss, and $\lambda_1, \lambda_2$ are the balancing hyperparamteres. We set $\lambda_1 = 1$ and $\lambda_2 = 0.5$ after a small hyperparameter search as illustrated in the Appendix D.6.

| Dataset | CIFAR10 | C10/SDXL | | | | |
|---|---|---|---|---|---|---|
| Ratio $P$ | 0% | 50% | 70% | 80% | 90% | 95% |
| ER | 63.93±2.40 | 60.40±1.54(-3.53) | 57.07±3.64(-6.86) | 54.18±3.42(-9.75) | 50.69±3.56(-13.24) | 47.39±2.76(-16.54) |
| DER++ | 64.31±2.63 | 60.24±2.02(-4.07) | 56.11±2.99(-8.20) | 51.62±3.28(-12.69) | 44.43±3.00(-19.88) | 40.46±2.92(-23.85) |
| ERACE | 60.19±2.51 | 56.17±2.08(-4.02) | 50.70±2.66(-9.49) | 46.86±4.61(-13.33) | 41.86±2.40(-18.33) | 37.56±2.46(-22.63) |
| OCM | 72.66±1.61 | 69.70±1.52(-2.96) | 66.68±1.69(-5.98) | 63.79±1.57(-8.87) | 60.41±1.36(-12.25) | 57.07±1.42(-15.59) |
| GSA | 66.91±1.57 | 63.47±1.89(-3.44) | 59.44±2.33(-7.47) | 56.60±2.63(-10.31) | 49.93±2.97(-16.98) | 45.77±2.60(-21.14) |
| OnPro | **74.87±1.58** | **72.46±1.36**(-2.41) | **68.79±1.17**(-6.08) | 66.07±1.23(-8.80) | 62.37±1.70(-12.50) | 56.41±2.59(-18.46) |
| ESRM | 67.35±1.14 | 67.95±0.88**(0.6)** | 67.47±1.43**(0.12)** | **66.81±1.59(-0.54)** | **64.59±1.59(-2.76)** | **60.04±2.29(-7.31)** |

| Dataset | CIFAR100 | C100/SDXL | | | | |
|---|---|---|---|---|---|---|
| Ratio $P$ | 0% | 50% | 70% | 80% | 90% | 95% |
| ER | 38.70±1.45 | 36.37±1.39(-2.33) | 32.82±1.62(-5.88) | 31.33±1.33(-7.37) | 27.09±1.02(-11.61) | 25.56±1.23(-13.14) |
| DER++ | 37.62±2.30 | 32.35±2.72(-5.27) | 28.32±2.45(-9.30) | 25.57±2.44(-12.05) | 19.56±1.88(-18.06) | 16.00±1.64(-21.62) |
| ERACE | 39.82±1.37 | 34.15±1.33(-5.67) | 28.16±1.37(-11.66) | 25.13±1.30(-14.69) | 19.37±1.49(-20.45) | 16.14±1.29(-23.68) |
| OCM | 42.01±1.07 | 40.21±1.15(-1.80) | 37.54±1.36(-4.47) | 34.78±1.32(-7.23) | 31.40±1.51(-10.61) | 28.84±1.28(-13.17) |
| GSA | 42.27±1.53 | 39.21±1.13(-3.06) | 35.21±1.62(-7.06) | 32.64±1.84(-9.63) | 27.62±1.15(-14.65) | 23.88±1.61(-18.39) |
| OnPro | 41.47±1.09 | 39.26±0.72(-2.21) | 35.64±0.60(-5.83) | 33.20±0.70(-8.27) | 30.20±0.84(-11.27) | 26.77±1.19(-14.70) |
| ESRM | **47.72±0.87** | **46.57±0.92(-1.15)** | **45.92±0.42(-1.80)** | **44.48±0.41(-3.24)** | **40.99±0.70(-6.73)** | **37.45±0.56(-10.27)** |

| Dataset | Tiny | Tiny/SDXL | | | | |
|---|---|---|---|---|---|---|
| Ratio $P$ | 0% | 50% | 70% | 80% | 90% | 95% |
| ER | 25.06±1.81 | 18.03±1.69(-7.03) | 13.59±1.86(-11.47) | 11.29±1.46(-13.77) | 6.38±0.89(-18.68) | 3.88±0.65(-21.18) |
| DER++ | 19.40±3.71 | 12.55±2.26(-6.85) | 9.71±1.41(-9.69) | 7.46±1.45(-11.94) | 4.49±0.83(-14.91) | 2.81±0.41**(-16.59)** |
| ERACE | 26.38±1.03 | 17.04±0.88(-9.34) | 11.23±0.69(-15.15) | 7.83±1.16(-18.55) | 4.09±0.58(-22.29) | 2.54±0.55(-23.84) |
| OCM | 31.94±1.44 | 25.21±0.65(-6.73) | 20.14±1.16(-11.80) | 16.16±0.64(-15.78) | 10.35±0.58(-21.59) | 5.37±0.63(-26.57) |
| GSA | 25.34±1.01 | 15.59±2.29(-9.75) | 12.55±1.65(-12.79) | 9.31±1.20(-16.03) | 5.95±0.83(-19.39) | 3.87±0.55(-21.47) |
| OnPro | 26.38±2.18 | 16.92±1.22(-9.46) | 13.23±1.26(-13.15) | 8.82±1.26(-17.56) | 4.80±1.12(-21.58) | 2.68±0.58(-23.70) |
| ESRM | **32.15±1.20** | **29.36±0.53(-2.79)** | **27.81±1.02(-4.34)** | **25.7±1.16(-6.45)** | **19.09±1.57(-13.06)** | **13.02±0.79**(-19.13) |

| Dataset | In-100 | In-100/SDXL | | | | |
|---|---|---|---|---|---|---|
| Ratio $P$ | 0% | 50% | 70% | 80% | 90% | 95% |
| ER | 33.35±1.84 | 30.49±0.91(-2.81) | 26.47±0.64(-6.83) | 23.61±0.68(-9.69) | 21.09±0.61(-12.21) | 18.54±0.92(-14.76) |
| DER++ | 34.89±2.27 | 29.98±4.35(-4.77) | 26.84±1.72(-7.91) | 23.72±2.10(-11.03) | 20.57±1.67(-14.18) | 17.52±2.06(-17.23) |
| ERACE | 38.43±1.45 | 32.96±1.55(-5.41) | 28.99±0.73(-9.38) | 25.28±0.77(-13.09) | 20.51±1.13(-17.86) | 16.91±0.64(-21.46) |
| OCM | 26.70±2.36 | 26.43±0.53(-0.27) | 23.70±1.11(-3.00) | 23.61±2.00(-3.09) | 22.21±1.09(-4.49) | 20.56±1.20**(-6.14)** |
| GSA | **40.85±1.04** | 37.35±1.23(-3.68) | 32.34±1.30(-8.69) | 29.47±0.44(-11.56) | 25.15±0.74(-15.88) | 21.89±0.25(-19.14) |
| OnPro | 38.47±1.13 | 36.17±2.70(-2.58) | 34.04±2.51(-4.71) | 33.01±1.07(-5.74) | 29.34±2.00(-9.41) | 25.14±1.26(-13.61) |
| ESRM | 39.72±1.05 | **39.64±0.76(-0.08)** | **39.62±0.90(-0.10)** | **39.53±0.83(-0.19)** | **36.58±0.80(-3.14)** | **32.86±0.99**(-6.86) |

Table 2: Average Accuracy (%; higher is better) on four benchmark datasets with different contamination ratios $P$. Numbers in parentheses indicate the performance degradation due to contamination compared to the clean setting. The average and deviation over five runs are reported for ImageNet-100 and 10 runs for other datasets.

# 6 Experiments

## 6.1 Experiment setup

**Datasets.** In the experiments, we used four benchmark datasets in evaluation, including CIFAR-10/100, TinyImageNet, and ImageNet-100. All of the datasets are split into tasks containing non-overlapping classes. The details about the task split are available in Appendix D.1.

**Baselines.** We evaluate the effectiveness of ESRM against six representative and state-of-the-art baselines, including ER [33], DER++ [5], ERACE [6], OCM [16], GSA [17], and OnPro [45].

**Implementation details.** We use full-width ResNet-18 (not pre-trained) as the backbone for all experiments. For a fair comparison, we conduct a hyperparameter search on CIFAR-100 (Memory Size = 5K) and apply the same hyperparameter to all settings. Stream batch size is set to 10 and memory batch size is set to 64. We do not use multiple updates trick for incoming batches as detailed in [2]. Detailed information about task allocation, hyperparameter search protocol, and data augmentation can be found in Appendix D.

**Buffer size.** For CIFAR-10 and CIFAR-100 experiments, the buffer size is set to 1,000 and 5,000, respectively. For the harder TinyImageNet experiments, the buffer size is set to 10,000. The buffer size of ImageNet-100 is set to 5,000. Appendix C.5 demonstrates more experiments with different memory buffer sizes.

| Dataset | In-100 | In-100/SDXL | | | | |
|---|---|---|---|---|---|---|
| Ratio $P$ | 0% | 50% | 70% | 80% | 90% | 95% |
| ER | 53.87±1.27 | 52.12±1.03 | 48.38±0.79 | 45.21±1.24 | 39.24±1.58 | 35.27±0.93 |
| DER++ | 61.14±2.38 | 56.99±1.85 | 53.56±2.86 | 50.26±2.06 | 44.08±1.89 | 37.92±2.19 |
| ERACE | 49.25±1.94 | 43.83±0.76 | 37.92±1.49 | 33.51±0.85 | 26.78±1.14 | 22.70±0.75 |
| OCM | 22.78±1.62 | 19.37±0.28 | 19.81±0.53 | 19.43±0.77 | 20.44±0.25 | 18.98±1.44 |
| GSA | 61.83±1.72 | 58.15±1.99 | 53.63±0.46 | 49.55±1.25 | 42.35±0.38 | 36.16±0.90 |
| OnPro | 39.28±0.84 | 38.98±3.16 | 40.07±1.72 | 38.45±2.80 | 36.45±1.51 | 32.94±1.33 |
| ESRM | **74.85±0.61** | **71.25±0.74** | **66.51±0.39** | **62.16±0.39** | **55.30±0.50** | **50.41±0.97** |

Table 3: Learning Accuracy (%; higher is better) on In-100/SDXL with various contamination ratio $P$.

| Dataset | In-100 | In-100/SDXL | | | | |
|---|---|---|---|---|---|---|
| Ratio $P$ | 0% | 50% | 70% | 80% | 90% | 95% |
| ER | 39.38±3.51 | 42.20±1.70 | 45.95±1.52 | 48.38±2.10 | 47.12±1.74 | 47.83±2.26 |
| DER++ | 42.26±5.72 | 48.35±8.20 | 51.00±5.01 | 53.73±4.78 | 54.28±5.20 | 54.93±6.77 |
| ERACE | 24.21±2.17 | 25.22±3.29 | 24.23±2.29 | 26.10±3.97 | 26.26±4.28 | 28.80±3.92 |
| OCM | **4.67±1.66** | **4.38±2.16** | **5.96±3.12** | **4.86±1.62** | **6.45±1.78** | **6.52±1.31** |
| GSA | 35.51±2.12 | 37.50±1.72 | 40.79±3.08 | 41.72±0.94 | 41.83±2.04 | 40.55±0.56 |
| OnPro | 15.96±2.23 | 23.12±4.69 | 22.87±3.14 | 23.89±1.70 | 24.72±1.80 | 28.17±2.93 |
| ESRM | 49.79±1.61 | 46.02±1.19 | 40.83±1.29 | 36.90±1.74 | 34.61±1.82 | 35.40±2.38 |

Table 4: Relative Forgetting (%; lower is better) on In-100/SDXL with various contamination ratio $P$.

| Memory strategy | $P = 70\%$ Acc. ↑ | $P = 80\%$ Acc. ↑ |
|---|---|---|
| Real Only (Idealized) | 47.43±0.62 | 46.85±0.76 |
| ES (Ours) | 45.92±0.42 | 44.48±0.41 |
| Random | 44.84±0.80 | 42.62±0.87 |
| Synthetic Only (Worst) | 25.91±0.79 | 24.74±0.91 |

Table 5: Comparison of different memory strategies on C100/SDXL dataset, with different contamination ratio $P$.

| Method | $P = 70\%$ Acc. ↑ | $P = 80\%$ Acc. ↑ |
|---|---|---|
| Baseline | 42.61±0.84 | 41.22±0.65 |
| w/o $\mathcal{L}_{SDC}$ | 44.30±0.78 | 42.59±0.69 |
| w/o $\mathcal{L}_{RM}$ | 44.03±0.73 | 42.32±0.52 |
| ESRM | 45.92±0.42 | 44.48±0.41 |

Table 6: Ablation of loss functions on C100/SDXL dataset, with different contamination ratio $P$. "Baseline" indicates ES+$\mathcal{L}_{CE}$.

## 6.2 Results and analysis

**Final Average Accuracy.** Table 2 shows the final average accuracy of learners trained with four datasets, including C10/SDXL, C100/SDXL, Tiny/SDXL, and In-100/SDXL, with different contamination ratios $P$. More results on C10/Mix, C100/Mix, and Tiny/Mix are given in Appendix C.3. For the six baselines, we can notice a significant performance drop when synthetic data contamination appears. Notably, when the contamination is severe (contamination ratio $P \geq 70\%$), the performance degradation is significant. Also, it can be observed that ESRM is less impacted by synthetic data contamination. For most datasets and contamination ratio $P$, the ESRM performance drop is the lowest of the compared methods. This is remarkably true for large values of $P$.

Apart from the robustness against synthetic data contamination, the absolute performance of ESRM is also attractive. In most cases, ESRM outperforms the baseline methods by a large margin. More interestingly, for some datasets, such as CIFAR-100 and ImageNet-100, even with an extreme contamination ratio, ESRM can still achieve a substantial performance, while the performance of most baseline methods is unsatisfactory.

**Plasticity and Stability Metrics.** We measure the model's plasticity and stability with Learning Accuracy (LA) [32] and Relative Forgetting (RF) [42], respectively. As shown in Table 3 and 4, for baseline methods, both plasticity and stability performance are hindered by synthetic data contamination. From the model plasticity perspective, ESRM alleviates the problem with a larger plasticity. For the model stability, ESRM solves the problem of stability degradation with the presence of contamination. It is notable that with RM loss and self-distillation loss, ESRM implicitly trades off some model stability in favor of plasticity. Plasticity and stability performance on other datasets are available in Appendix C.4.

**Domain-Incremental Learning Results.** While Class-Incremental Learning (CIL) setting is the standard evaluation protocol in online CL, we also evaluated the performance of ESRM in Domain-Incremental Learning (DIL) scenarios. We conducted the experiment with the 20 coarse labels of the

| Dataset | DIL-CIFAR20 | DIL-C20/SDXL | | |
|---------|-------------|--------------|---|---|
| Ratio $P$ | 0% | 50% | 70% | 80% |
| ER | $53.56_{\pm1.42}$ | $51.91_{\pm1.63}(-1.65)$ | $49.61_{\pm0.76}(-3.95)$ | $47.25_{\pm0.73}(-6.31)$ |
| DER++ | $56.43_{\pm0.65}$ | $52.46_{\pm1.19}(-3.97)$ | $48.73_{\pm1.50}(-7.70)$ | $44.60_{\pm1.79}(-11.83)$ |
| OCM | $56.02_{\pm1.17}$ | $53.87_{\pm0.57}(-2.15)$ | $52.69_{\pm0.93}(-3.33)$ | $50.57_{\pm0.80}(-5.45)$ |
| GSA | $53.67_{\pm2.60}$ | $51.54_{\pm1.76}(-2.13)$ | $47.16_{\pm1.76}(-6.51)$ | $45.55_{\pm1.17}(-8.12)$ |
| OnPro | $31.81_{\pm1.21}$ | $30.43_{\pm0.72}(-1.38)$ | $29.19_{\pm0.82}(-2.62)$ | $27.79_{\pm0.86}\mathbf{(-4.02)}$ |
| Ours | $\mathbf{64.27_{\pm0.46}}$ | $\mathbf{63.35_{\pm0.70}(-0.92)}$ | $\mathbf{61.86_{\pm0.54}(-2.41)}$ | $\mathbf{59.94_{\pm0.71}}(-4.33)$ |

Table 7: Final Average Accuracy (%; higher is better) on DIL-C20/SDXL dataset. Numbers in parentheses indicate the performance degradation due to synthetic contamination compared to the clean setting. The average and deviation over 10 runs are reported.

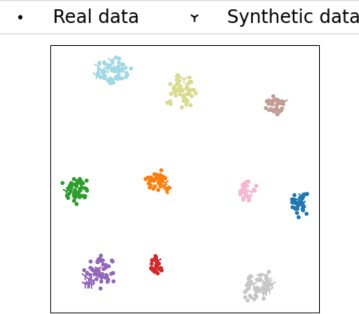

Figure 5: T-SNE visualization of memory data produced by ESRM at the end of training on the In-100/SDXL ($P = 50\%$) dataset. For clarity, only the first 10 classes are visualized.

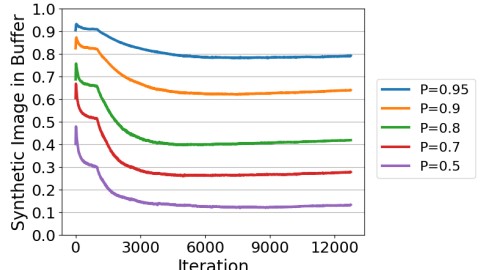

Figure 6: The percentage of synthetic data in the memory buffer throughout the training of ESRM on the In-100/SDXL dataset with different contamination ratios ($P$). The average value of 5 runs is plotted.

CIFAR-100 dataset. Since the 100 classes in CIFAR-100 are grouped into 20 superclasses with 5 fine-grained classes for each superclass, we split the CIFAR-100 dataset with 5 domain increment steps. For each step, we feed the model with the training data of a fine-grained class from each superclass. Because the model only classifies 20 coarse labels, we refer to this dataset as DIL-CIFAR20. Similar to the simulated CIFAR100/SDXL dataset, we replace the images in the DIL-CIFAR20 dataset with its Stable Diffusion XL generated counterpart with a contamination ratio P, as per the protocol in Sec. 3.2.

Table 7 shows the final average accuracy with different contamination ratios. Notably, we adapted the CIL-specifically designed components in OnPro and GSA to the DIL scenario, and the performance suffered a decent loss. We did not report ERACE results because its Asymmetric Cross Entropy (ACE) loss converges to standard cross-entropy loss in the DIL scenario, making it equivalent to vanilla ER. The experimental results show that ESRM can yield robust performance against domain shift in the DIL setting, under different synthetic contamination situations, which validates the efficiency of ESRM under DIL settings.

### 6.3 Ablation studies

**Effect of ES.** To evaluate the effect of ES, we substitute ES with three different memory strategies: random sampling, storing real data only, and storing synthetic data only. Note that storing real or synthetic data requires knowing the ground truth of an image's synthetic status, which is not practical in realistic settings. Additionally, storing only real data is an idealized case for memory management, while storing only synthetic data represents the worst-case scenario. As shown in Table 5, both ES and random reservoir sampling [21, 40] outperform the worst-case scenario by a large margin. Moreover, ES outperforms the random sampling significantly.

**Effects of loss terms.** We also conduct experiments to verify the effects of loss terms in Eq. 5. As shown in Table 6, Both $\mathcal{L}_{SDC}$ and $\mathcal{L}_{RM}$ can benefit the final average accuracy of the classification. Furthermore, the combination of the two loss terms can further improve the final accuracy, validating that both terms complement each other.

# 7 Discussions

**The alleviation of feature misalignment.** As mentioned in Obs. 4, baseline methods suffer performance degradation due to the misalignment between the inferior feature embedding of real images and the superior feature embedding of synthetic samples. Fig. 5 presents the t-SNE visualization of memory data at the end of training of ESRM on the In-100/SDXL dataset. Similar to Fig. 3, only the first 10 classes are visualized for clarity. Compared to ER and OnPro, the embeddings of synthetic and real samples in ESRM are better aligned, facilitated by the RM.

**Training dynamics of memory buffer.** To intuitively demonstrate the effect of ES, we visualized the percentage of synthetic data in the memory buffer throughout the whole training process. Fig. 6 displays the curve of the percentage of synthetic data in the memory buffer when the model is trained with ESRM on In-100/SDXL with different contamination ratios $P$. To generate this curve, we checked the memory buffer every 10 iterations. As shown in the figure, the percentage of synthetic data is close to the contamination ratio $P$ in the early stages of training. As training progresses, the amount of synthetic data decreases. This trend intuitively illustrates the effect of ES in selecting real samples. Furthermore, we take a model trained with ESRM at the end of training and use the model's entropy

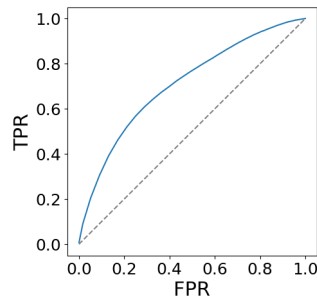

Figure 7: The ROC curve of the model trained with ESRM on the In-100/SDXL dataset ($P = 50\%$) in predicting the synthetic status of samples in the training dataset. Real samples are regarded as positives and synthetic samples as negatives.

as the criterion to categorize the synthetic status of samples in the training dataset and generate an ROC curve, as shown in Fig. 7. We regard real data as positive and use models trained with ESRM on In-100/SDXL ($P = 50\%$) to categorize the samples in the training set. The AUC of the ROC curve is 0.7098, showing the effect of the entropy criterion in discriminating real and synthetic samples.

# 8 Conclusion

With the widespread availability of advanced generative models, the prevalence of AI-generated images appears inevitable, posing a potential challenge for researchers attempting to collect datasets devoid of AI-generated content from the Internet. In this paper, we examine the potential side effects of AI-powered image generation on the continual learning community. First, we experimentally demonstrate that synthetic data has become a potential source of data pollution. We spot a catastrophic performance loss when the contaminated datasets are used to train continual learning models. Based on our experiments, we identify and summarize four typical characteristics of synthetic data when involved in the training of continual learners. Additionally, we propose ESRM, a method designed to alleviate performance deterioration, maintaining satisfactory performance even with highly contaminated datasets. Lastly, we hope our work highlights the need for improved regulation and systematic control over generated data, such as watermarking AI-generated content before publication. Internet data is a valuable resource accumulated over decades. We believe ensuring the integrity of Internet data is crucial for the future soundness of AI development.

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

**Algorithm 1** PyTorch-like pseudo-code of ES.

```
# model: continual learning model.
# criteria(): loss function as in Eq. 5.
# n_seen_so_far: count of images seen by the buffer.
# random(): function returns random values in (0,1).
# buffer_labels: 1D Tensor of size [buffer_size] storing the labels.
# buffer_ent: 1D Tensor of size [buffer_size] storing the entropy values.
# update_ent(): Update all entropy values in the buffer.

n_seen_so_far = 0
for t in tasks:
    for img, label in dataloader:
        # train the network with stream data
        model.train()
        mem_img, mem_label = mem_sample()
        c_img, c_label = concat((img, mem_img), (label, mem_label)) # combined batch
        loss = criteria(model(aug(c_img)), c_label) # Eq. 5.
        loss.backward()
        optimizer.step()

        # ES updates
        # calculate the entropy criteria of stream data
        model.eval()
        logits = model(img)
        prob = softmax(logits)
        entropy = -torch.sum(prob * torch.log(prob))

        # update buffer
        threshold = torch.quantile(entropy, 0.5) # Step 1 in ES
        stream_img = img[entropy > threshold]
        stream_label = label[entropy > threshold]
        stream_entropy = entropy[entropy > threshold]

        for x, y, ent in zip(stream_img, stream_label, stream_ent):
            nominate = int(random()*(n_seen_so_far + 1)) # Step 2 in ES
            if n_seen_so_far < buffer_size: # if the buffer is not full
                nominate = n_seen_so_far
                replace_data(nominate, x, y, ent)
                n_seen_so_far += 1
            elif nominate < buffer_size:
                nominate_class = buffer_labels[nominate] # Step 3 in ES
                idx = buffer_ent[buffer_labels == nominate_class].argmin()
                replace_data(idx, x, y, ent) # Step 4 in ES
                n_seen_so_far += 1

    update_ent()
```

## A   Limitations

The paper investigates the potential impact of synthetic data contamination on online CL research and proposes a method to alleviate side effects caused by the contamination. Nevertheless, our research has some limitations. Firstly, in the evaluation, we only use five generative models, including Stable Diffusion v1.4, Stable Diffusion v2.1, Stable Diffusion XL, VQDM, and GLIDE. Other excellent commercial generative works, such as Midjourney and DALL-E, are not included in the data generation. With limited computation/resources, we could not exhaust all generative methods.

Secondly, the way we produce the generated dataset is simple: we use prompts like "an image of a <class_name>" in the generation. In reality, the users' prompts are usually more diverse. Some works use LLMs to simulate more realistic prompts, where we leave a further in-depth analysis for future research.

## B   Pseudo code for ES

The PyTorch-like pseudo code showing how the ES updates the memory buffer is shown in Alg. 1.

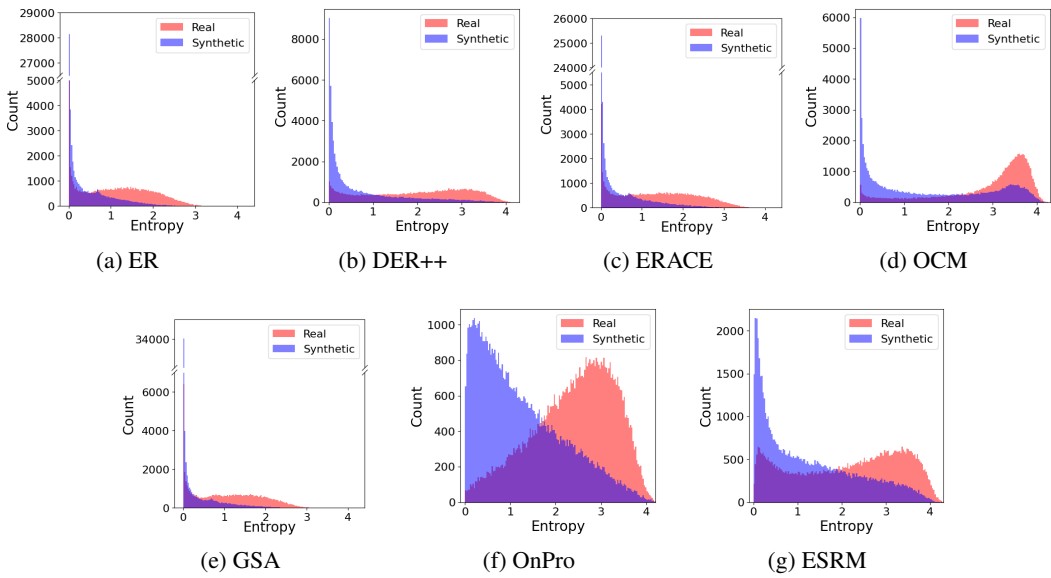

Figure 8: The entropy distribution of the training set produced by all methods on In-100/SDXL $P = 50\%$ at the end of the training.

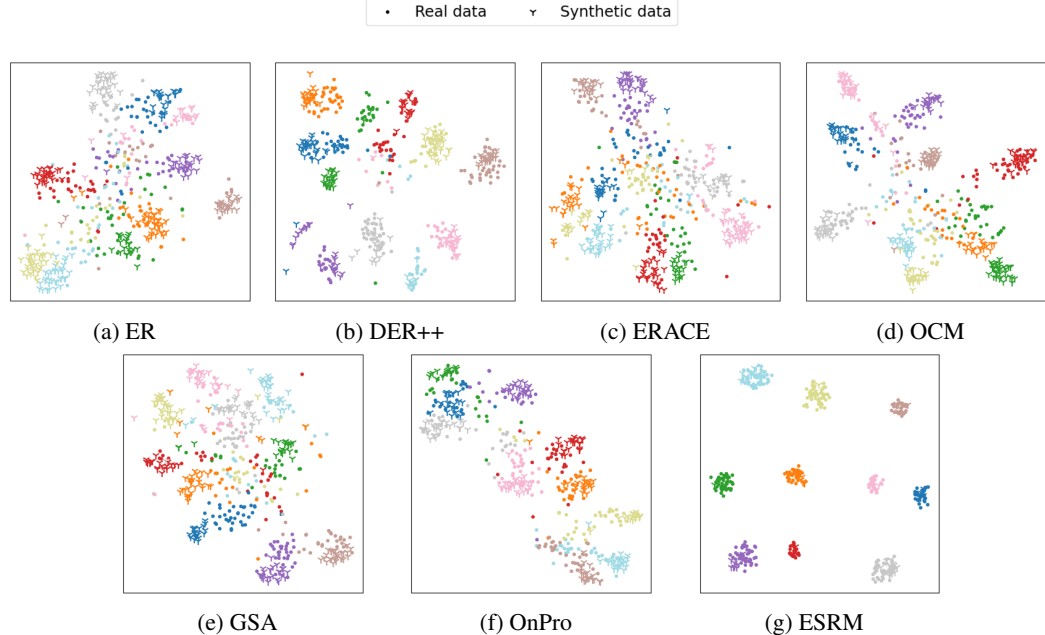

Figure 9: T-SNE visualization of the memory data at the end of training on In-100/SDXL ($P = 50\%$). For clarity, only the first 10 classes are visualized.

## C    Extra Experiments

### C.1    Entropy distribution of other baselines

As mentioned in Sec. 4, we show the entropy distribution produced by other baselines, when trained on In-100/SDXL ($P = 50\%$). Fig. 8 illustrates the entropy distribution histogram, which is calculated at the end of the training on the whole contaminated dataset. Similar to the result in Fig. 2, the entropy distribution of the synthetic data is saliently lower than the entropy distribution of the real data.

| Dataset | CIFAR10 | C10/Mix | | | | |
|---|---|---|---|---|---|---|
| Ratio $P$ | 0% | 50% | 70% | 80% | 90% | 95% |
| ER | 63.93±2.40 | 61.74±2.36(-2.19) | 57.54±2.04(-6.39) | 57.18±2.51(-6.75) | 53.71±2.32(-10.22) | 52.21±2.92(-11.72) |
| DER++ | 64.31±2.63 | 61.62±2.57(-2.69) | 59.64±4.24(-4.67) | 57.31±3.36(-7.00) | 53.28±4.10(-11.03) | 51.68±3.16(-12.63) |
| ERACE | 60.19±2.51 | 58.21±2.38(-1.98) | 55.01±2.59(-5.18) | 51.15±3.45(-9.04) | 49.16±3.10(-11.03) | 47.74±2.45(-12.45) |
| OCM | 72.66±1.61 | 70.13±1.38(-2.53) | 66.62±1.69(-6.04) | 65.08±1.63(-7.58) | 62.86±1.39(-9.80) | 61.60±1.99(-11.06) |
| GSA | 66.91±1.57 | 63.05±2.02(-3.86) | 62.45±1.52(-4.46) | 60.94±1.50(-5.97) | 57.51±2.58(-9.40) | 55.84±1.43(-11.07) |
| OnPro | **74.87±1.58** | **72.45±1.97**(-2.42) | **69.79±1.78**(-5.08) | **67.13±1.41**(-7.74) | 64.23±1.17(-10.64) | 63.63±0.85(-11.24) |
| ESRM | 67.35±1.14 | 68.04±1.33(**0.69**) | 67.60±1.16(**0.25**) | 66.73±1.03(**-0.62**) | **64.74±0.88(-2.61)** | **64.38±1.60(-2.97)** |

| Dataset | CIFAR100 | C100/Mix | | | | |
|---|---|---|---|---|---|---|
| Ratio $P$ | 0% | 50% | 70% | 80% | 90% | 95% |
| ER | 38.70±1.45 | 36.54±1.40(-2.16) | 33.82±1.13(-4.88) | 32.33±0.77(-6.37) | 30.24±0.80(-8.46) | 28.36±1.22(-10.34) |
| DER++ | 37.62±2.30 | 35.38±2.65(-2.24) | 31.58±1.91(-6.04) | 29.71±2.19(-7.91) | 26.75±1.51(-10.87) | 24.78±1.17(-12.84) |
| ERACE | 39.82±1.37 | 35.65±1.04(-4.17) | 31.44±2.28(-8.38) | 29.21±1.30(-10.61) | 26.24±1.17(-13.58) | 23.42±1.41(-16.4) |
| OCM | 42.01±1.07 | 39.78±0.94(-2.23) | 37.19±1.19(-4.82) | 35.51±1.58(-6.50) | 33.37±1.04(-8.64) | 32.01±0.63(-10.00) |
| GSA | 42.27±1.53 | 39.88±1.54(-2.39) | 36.72±0.68(-5.55) | 34.27±1.59(-8.00) | 31.83±1.42(-10.44) | 30.17±1.61(-12.10) |
| OnPro | 41.47±1.09 | 38.78±1.27(-2.69) | 36.53±0.63(-4.94) | 34.48±0.58(-6.99) | 32.08±1.07(-9.39) | 31.06±1.71(-10.41) |
| ESRM | **47.72±0.87** | **45.58±0.88(-2.14)** | **44.02±0.65(-3.70)** | **43.81±0.95(-3.91)** | **41.10±0.71(-6.62)** | **38.13±0.48(-9.59)** |

| Dataset | Tiny | Tiny/Mix | | | | |
|---|---|---|---|---|---|---|
| Ratio $P$ | 0% | 50% | 70% | 80% | 90% | 95% |
| ER | 25.06±1.81 | 17.28±1.88(-7.78) | 13.52±1.49(-11.54) | 10.26±1.40(-14.80) | 6.73±0.96(-18.33) | 3.46±0.68(-21.60) |
| DER++ | 19.40±3.71 | 13.86±2.44(-5.54) | 10.56±0.72(-8.84) | 7.35±1.78(-12.05) | 3.66±0.61(-15.74) | 1.59±0.46(**-17.81**) |
| ERACE | 26.38±1.03 | 18.70±1.23(-7.68) | 13.99±0.81(-12.39) | 10.35±1.03(-16.03) | 5.62±0.67(-20.76) | 3.01±0.47(-23.37) |
| OCM | 31.94±1.44 | 24.14±1.17(-7.80) | 19.18±1.18(-12.76) | 15.06±0.83(-16.88) | 8.82±0.93(-23.12) | 4.28±0.76(-27.66) |
| GSA | 25.34±1.01 | 17.11±1.40(-8.23) | 13.49±1.65(-11.85) | 10.02±0.96(-15.32) | 6.00±0.86(-19.34) | 3.26±0.68(-22.08) |
| OnPro | 26.38±2.18 | 17.03±1.67(-9.35) | 11.84±1.42(-14.54) | 8.82±1.24(-17.56) | 3.79±0.61(-22.59) | 1.72±0.62(-24.66) |
| ESRM | **32.15±1.20** | **27.37±1.24(-4.78)** | **25.65±1.12(-6.50)** | **23.52±1.24(-8.63)** | **17.29±1.16(-14.86)** | **11.50±0.74(-20.65)** |

Table 8: Average Accuracy (%; higher is better) on four benchmark datasets with different contamination ratios $P$. Numbers in parentheses indicate the performance degradation due to contamination compared to the clean setting. The average and deviation over 10 runs are reported.

## C.2 T-SNE visualization of other baselines

We show additional experiments of t-SNE visualization of memory data produced by other baseline methods. As shown in Fig. 9, we visualize the memory embeddings of different baselines on the In-100/SDXL ($P = 50\%$) dataset. For clarity, we only visualize the first 10 classes. Similar to the results in Fig. 3, the synthetic data are better clustered compared with the real data. This proves the Obs. 4.

## C.3 Performance on C10/Mix, C100/Mix and Tiny/Mix Dataset

As mentioned in Sec. 6, we would like to include the experiment results on C10/Mix, C100/Mix, and Tiny/Mix datasets. The final average accuracy of different methods on the dataset is included in Table 8. Similar to the results in Table 2, ESRM has less performance degradation and better performance in most cases.

## C.4 Plasticity and stability performance on other datasets

As mentioned in Sec. 6.2, the plasticity and stability metrics of methods on other datasets are demonstrated in Table 9 and 10. Similar to the results in Table 3 and 4, in most settings, the plasticity metric (LA) and stability metric (RF) of baseline methods drop with an increased contamination ratio $P$. For the model plasticity, ESRM alleviates the issue with a larger plasticity. From a stability perspective, ESRM addresses the issue of stability degradation with the presence of contamination.

## C.5 The impact of buffer size.

In Sec. 6.2, we evaluate the effectiveness of ESRM in a limited buffer size setting. To evaluate our methods' scalability against different buffer sizes $M$ and contamination ratio $P$, we compare the accuracy of different methods on C100/SDXL with different buffer sizes $M$, as shown in Table 11. Similar to the results in Table 2, ESRM can obtain better performance when the dataset is contaminated with synthetic data.

| Dataset | CIFAR10 | C10/SDXL | | | | |
|---|---|---|---|---|---|---|
| Ratio $P$ | 0% | 50% | 70% | 80% | 90% | 95% |
| ER | $80.05_{\pm3.24}$ | $79.56_{\pm2.51}$ | $74.76_{\pm3.04}$ | $74.15_{\pm3.80}$ | $72.69_{\pm3.02}$ | $69.60_{\pm4.38}$ |
| DER++ | $78.35_{\pm1.88}$ | $74.51_{\pm3.63}$ | $71.33_{\pm4.16}$ | $67.68_{\pm4.23}$ | $62.96_{\pm2.89}$ | $60.45_{\pm3.05}$ |
| ERACE | $60.64_{\pm3.24}$ | $57.16_{\pm3.18}$ | $49.61_{\pm5.13}$ | $46.44_{\pm5.47}$ | $43.70_{\pm3.48}$ | $40.94_{\pm4.11}$ |
| OCM | $79.84_{\pm3.01}$ | $79.25_{\pm2.28}$ | $76.43_{\pm3.26}$ | $76.03_{\pm3.39}$ | $74.41_{\pm2.51}$ | $71.11_{\pm1.94}$ |
| GSA | $77.28_{\pm3.11}$ | $74.72_{\pm3.39}$ | $69.10_{\pm2.65}$ | $66.51_{\pm3.92}$ | $60.17_{\pm4.53}$ | $56.57_{\pm7.26}$ |
| OnPro | $84.87_{\pm2.55}$ | $83.12_{\pm3.30}$ | $82.98_{\pm1.75}$ | $79.81_{\pm2.82}$ | $77.99_{\pm2.88}$ | $73.39_{\pm3.26}$ |
| ESRM | $91.36_{\pm1.25}$ | $88.19_{\pm1.05}$ | $85.57_{\pm1.59}$ | $83.32_{\pm1.70}$ | $79.48_{\pm2.09}$ | $76.09_{\pm3.10}$ |

| Dataset | CIFAR100 | C100/SDXL | | | | |
|---|---|---|---|---|---|---|
| Ratio $P$ | 0% | 50% | 70% | 80% | 90% | 95% |
| ER | $50.47_{\pm1.31}$ | $48.60_{\pm1.77}$ | $46.08_{\pm1.69}$ | $43.19_{\pm1.28}$ | $39.66_{\pm1.47}$ | $37.21_{\pm1.49}$ |
| DER++ | $55.91_{\pm3.80}$ | $50.80_{\pm2.52}$ | $46.42_{\pm2.86}$ | $43.17_{\pm2.72}$ | $36.79_{\pm3.14}$ | $31.44_{\pm2.20}$ |
| ERACE | $41.29_{\pm1.72}$ | $35.83_{\pm0.47}$ | $30.06_{\pm1.22}$ | $26.56_{\pm1.24}$ | $21.70_{\pm1.64}$ | $18.10_{\pm0.68}$ |
| OCM | $43.56_{\pm1.56}$ | $43.30_{\pm1.86}$ | $40.59_{\pm1.52}$ | $38.62_{\pm1.94}$ | $35.73_{\pm1.57}$ | $33.93_{\pm1.81}$ |
| GSA | $50.76_{\pm1.64}$ | $48.30_{\pm2.28}$ | $43.70_{\pm2.17}$ | $40.49_{\pm1.75}$ | $34.29_{\pm1.48}$ | $30.92_{\pm1.56}$ |
| OnPro | $41.98_{\pm1.44}$ | $41.58_{\pm1.85}$ | $39.36_{\pm1.36}$ | $36.99_{\pm1.87}$ | $33.95_{\pm1.61}$ | $30.97_{\pm1.81}$ |
| ESRM | $70.85_{\pm0.95}$ | $67.07_{\pm0.77}$ | $62.83_{\pm0.64}$ | $59.45_{\pm1.00}$ | $54.33_{\pm0.52}$ | $49.89_{\pm0.78}$ |

| Dataset | Tiny | Tiny/SDXL | | | | |
|---|---|---|---|---|---|---|
| Ratio $P$ | 0% | 50% | 70% | 80% | 90% | 95% |
| ER | $64.44_{\pm1.27}$ | $56.76_{\pm1.23}$ | $51.90_{\pm1.95}$ | $47.26_{\pm0.94}$ | $36.63_{\pm1.39}$ | $24.89_{\pm1.23}$ |
| DER++ | $70.28_{\pm2.42}$ | $64.84_{\pm1.48}$ | $61.25_{\pm2.79}$ | $58.82_{\pm1.92}$ | $54.52_{\pm1.88}$ | $49.59_{\pm1.56}$ |
| ERACE | $4.60_{\pm0.88}$ | $3.50_{\pm0.45}$ | $3.00_{\pm0.50}$ | $2.80_{\pm0.68}$ | $2.24_{\pm0.35}$ | $1.96_{\pm0.34}$ |
| OCM | $14.91_{\pm2.23}$ | $9.34_{\pm1.27}$ | $7.49_{\pm1.12}$ | $6.65_{\pm0.94}$ | $5.43_{\pm1.04}$ | $4.56_{\pm0.89}$ |
| GSA | $14.95_{\pm0.52}$ | $8.67_{\pm1.38}$ | $6.68_{\pm0.62}$ | $5.06_{\pm0.61}$ | $3.19_{\pm0.82}$ | $2.37_{\pm0.36}$ |
| OnPro | $15.82_{\pm1.04}$ | $11.18_{\pm1.40}$ | $9.45_{\pm1.62}$ | $8.68_{\pm0.72}$ | $8.30_{\pm1.30}$ | $6.75_{\pm1.36}$ |
| ESRM | $81.47_{\pm0.58}$ | $73.76_{\pm0.44}$ | $66.65_{\pm0.64}$ | $60.88_{\pm0.96}$ | $50.58_{\pm1.15}$ | $38.43_{\pm0.91}$ |

| Dataset | CIFAR10 | C10/Mix | | | | |
|---|---|---|---|---|---|---|
| Ratio $P$ | 0% | 50% | 70% | 80% | 90% | 95% |
| ER | $80.05_{\pm3.24}$ | $77.03_{\pm3.53}$ | $75.06_{\pm3.60}$ | $74.57_{\pm3.31}$ | $72.41_{\pm3.11}$ | $71.68_{\pm3.89}$ |
| DER++ | $78.35_{\pm1.88}$ | $75.53_{\pm2.35}$ | $74.16_{\pm3.44}$ | $70.95_{\pm2.93}$ | $69.57_{\pm2.82}$ | $67.31_{\pm2.27}$ |
| ERACE | $60.64_{\pm3.24}$ | $57.57_{\pm4.67}$ | $52.87_{\pm3.40}$ | $49.89_{\pm4.71}$ | $49.55_{\pm4.95}$ | $46.53_{\pm4.00}$ |
| OCM | $79.84_{\pm3.01}$ | $78.75_{\pm3.58}$ | $76.91_{\pm2.67}$ | $75.78_{\pm2.84}$ | $73.60_{\pm2.72}$ | $72.36_{\pm2.44}$ |
| GSA | $77.28_{\pm3.11}$ | $72.38_{\pm2.99}$ | $72.11_{\pm3.74}$ | $69.66_{\pm2.58}$ | $64.43_{\pm4.81}$ | $60.80_{\pm4.32}$ |
| OnPro | $84.87_{\pm2.55}$ | $82.96_{\pm2.97}$ | $80.94_{\pm2.79}$ | $79.00_{\pm2.31}$ | $76.94_{\pm2.75}$ | $76.17_{\pm3.31}$ |
| ESRM | $91.36_{\pm1.25}$ | $88.81_{\pm1.49}$ | $85.98_{\pm1.44}$ | $84.60_{\pm1.96}$ | $82.62_{\pm0.82}$ | $81.11_{\pm1.58}$ |

| Dataset | CIFAR100 | C100/Mix | | | | |
|---|---|---|---|---|---|---|
| Ratio $P$ | 0% | 50% | 70% | 80% | 90% | 95% |
| ER | $50.47_{\pm1.31}$ | $48.64_{\pm1.91}$ | $45.45_{\pm1.37}$ | $43.30_{\pm1.20}$ | $41.36_{\pm1.29}$ | $39.28_{\pm1.74}$ |
| DER++ | $55.91_{\pm3.80}$ | $54.30_{\pm3.50}$ | $50.02_{\pm1.74}$ | $47.44_{\pm2.68}$ | $43.39_{\pm1.61}$ | $41.68_{\pm2.21}$ |
| ERACE | $41.29_{\pm1.72}$ | $37.41_{\pm1.00}$ | $33.96_{\pm1.31}$ | $31.89_{\pm1.16}$ | $28.65_{\pm1.07}$ | $26.62_{\pm0.89}$ |
| OCM | $43.56_{\pm1.56}$ | $42.41_{\pm2.05}$ | $39.26_{\pm1.65}$ | $38.16_{\pm1.90}$ | $37.45_{\pm1.48}$ | $36.16_{\pm1.47}$ |
| GSA | $50.76_{\pm1.64}$ | $48.07_{\pm1.31}$ | $45.37_{\pm1.44}$ | $42.83_{\pm1.59}$ | $39.65_{\pm1.21}$ | $37.85_{\pm1.41}$ |
| OnPro | $41.98_{\pm1.44}$ | $41.62_{\pm1.46}$ | $39.24_{\pm1.53}$ | $36.95_{\pm1.60}$ | $36.02_{\pm1.01}$ | $34.94_{\pm1.11}$ |
| ESRM | $70.85_{\pm0.95}$ | $67.53_{\pm1.08}$ | $64.04_{\pm0.91}$ | $61.34_{\pm1.01}$ | $57.57_{\pm0.73}$ | $54.76_{\pm0.75}$ |

| Dataset | Tiny | Tiny/Mix | | | | |
|---|---|---|---|---|---|---|
| Ratio $P$ | 0% | 50% | 70% | 80% | 90% | 95% |
| ER | $64.44_{\pm1.27}$ | $55.16_{\pm1.06}$ | $50.39_{\pm2.59}$ | $44.24_{\pm2.08}$ | $32.22_{\pm1.74}$ | $22.06_{\pm1.74}$ |
| DER++ | $70.28_{\pm2.42}$ | $64.64_{\pm1.19}$ | $62.77_{\pm2.16}$ | $60.53_{\pm2.12}$ | $53.28_{\pm2.80}$ | $46.38_{\pm2.73}$ |
| ERACE | $4.60_{\pm0.88}$ | $3.04_{\pm0.75}$ | $2.63_{\pm0.55}$ | $2.29_{\pm0.48}$ | $1.87_{\pm0.37}$ | $1.86_{\pm0.25}$ |
| OCM | $14.91_{\pm2.23}$ | $8.58_{\pm1.20}$ | $6.18_{\pm1.17}$ | $4.72_{\pm1.03}$ | $4.01_{\pm0.75}$ | $3.42_{\pm0.63}$ |
| GSA | $14.95_{\pm0.52}$ | $8.66_{\pm1.18}$ | $6.53_{\pm1.00}$ | $4.90_{\pm0.66}$ | $2.84_{\pm0.58}$ | $2.06_{\pm0.49}$ |
| OnPro | $15.82_{\pm1.04}$ | $9.83_{\pm1.06}$ | $7.92_{\pm1.22}$ | $6.89_{\pm1.13}$ | $5.62_{\pm0.85}$ | $4.32_{\pm1.28}$ |
| ESRM | $81.47_{\pm0.58}$ | $74.50_{\pm0.59}$ | $66.60_{\pm1.00}$ | $60.56_{\pm0.75}$ | $48.22_{\pm1.16}$ | $34.33_{\pm1.12}$ |

Table 9: Learning Accuracy (%; higher is better) on contaminated datasets with various contamination ratio $P$.

| Dataset | CIFAR10 | C10/SDXL | | | | |
|---|---|---|---|---|---|---|
| Ratio $P$ | 0% | 50% | 70% | 80% | 90% | 95% |
| ER | 20.30$\pm$4.50 | 23.47$\pm$3.80 | 23.56$\pm$5.66 | 25.71$\pm$5.14 | 29.74$\pm$5.20 | 30.08$\pm$6.21 |
| DER++ | 17.36$\pm$2.62 | 18.64$\pm$3.57 | 19.65$\pm$2.64 | 21.46$\pm$3.63 | 25.74$\pm$4.63 | 28.72$\pm$4.44 |
| ERACE | 9.59$\pm$2.12 | 10.69$\pm$1.30 | 13.97$\pm$3.22 | 15.83$\pm$4.33 | 15.70$\pm$3.72 | 20.51$\pm$2.05 |
| OCM | 10.43$\pm$2.51 | 12.44$\pm$1.72 | 13.43$\pm$3.90 | 16.39$\pm$3.01 | 18.21$\pm$2.95 | 18.68$\pm$3.39 |
| GSA | 16.25$\pm$2.62 | 17.41$\pm$2.92 | 16.50$\pm$3.44 | 17.10$\pm$4.02 | 21.34$\pm$5.05 | 21.12$\pm$7.01 |
| OnPro | 12.07$\pm$3.19 | 13.05$\pm$2.64 | 15.65$\pm$2.96 | 16.88$\pm$2.16 | 19.12$\pm$4.23 | 21.94$\pm$5.57 |
| ESRM | 26.12$\pm$2.05 | 23.03$\pm$2.16 | 21.27$\pm$1.96 | 20.24$\pm$2.42 | 20.52$\pm$2.70 | 23.99$\pm$3.23 |

| Dataset | CIFAR100 | C100/SDXL | | | | |
|---|---|---|---|---|---|---|
| Ratio $P$ | 0% | 50% | 70% | 80% | 90% | 95% |
| ER | 23.86$\pm$2.73 | 26.06$\pm$4.53 | 28.51$\pm$3.56 | 27.34$\pm$3.41 | 31.48$\pm$2.95 | 30.91$\pm$3.12 |
| DER++ | 33.11$\pm$4.58 | 35.73$\pm$5.20 | 38.56$\pm$4.77 | 39.97$\pm$5.72 | 46.20$\pm$4.12 | 47.89$\pm$5.86 |
| ERACE | 9.81$\pm$1.72 | 12.26$\pm$1.91 | 13.86$\pm$2.75 | 14.15$\pm$3.64 | 18.54$\pm$5.33 | 19.61$\pm$5.17 |
| OCM | 8.35$\pm$2.07 | 10.46$\pm$2.62 | 10.93$\pm$3.57 | 12.61$\pm$1.80 | 15.41$\pm$2.48 | 17.60$\pm$3.00 |
| GSA | 19.62$\pm$2.73 | 21.56$\pm$3.38 | 22.69$\pm$2.95 | 23.05$\pm$4.13 | 23.45$\pm$3.00 | 25.38$\pm$4.53 |
| OnPro | 11.91$\pm$1.87 | 13.28$\pm$2.40 | 15.99$\pm$1.99 | 16.58$\pm$2.13 | 17.01$\pm$2.50 | 18.78$\pm$3.58 |
| ESRM | 32.67$\pm$1.46 | 30.75$\pm$1.48 | 26.93$\pm$1.33 | 25.24$\pm$1.64 | 24.53$\pm$1.04 | 24.85$\pm$1.77 |

| Dataset | Tiny | Tiny/SDXL | | | | |
|---|---|---|---|---|---|---|
| Ratio $P$ | 0% | 50% | 70% | 80% | 90% | 95% |
| ER | 61.84$\pm$2.65 | 69.08$\pm$3.09 | 74.91$\pm$3.41 | 76.94$\pm$2.89 | 83.45$\pm$2.34 | 86.34$\pm$2.27 |
| DER++ | 72.51$\pm$5.53 | 80.83$\pm$3.51 | 84.35$\pm$2.27 | 87.53$\pm$2.58 | 91.88$\pm$2.09 | 94.46$\pm$0.72 |
| ERACE | 36.40$\pm$2.74 | 44.92$\pm$2.49 | 54.44$\pm$3.07 | 62.57$\pm$5.27 | 76.19$\pm$3.48 | 78.71$\pm$4.21 |
| OCM | 32.25$\pm$1.44 | 37.15$\pm$1.73 | 43.82$\pm$2.82 | 49.69$\pm$2.54 | 58.53$\pm$2.89 | 73.77$\pm$4.01 |
| GSA | 44.78$\pm$2.76 | 55.72$\pm$5.38 | 58.39$\pm$3.46 | 64.32$\pm$3.04 | 71.39$\pm$3.42 | 76.41$\pm$4.15 |
| OnPro | 42.81$\pm$4.63 | 52.63$\pm$4.17 | 56.83$\pm$3.74 | 65.18$\pm$3.77 | 75.11$\pm$5.65 | 80.98$\pm$1.55 |
| ESRM | 61.53$\pm$1.36 | 61.35$\pm$0.66 | 59.79$\pm$1.82 | 59.55$\pm$2.17 | 65.44$\pm$2.72 | 71.04$\pm$2.10 |

| Dataset | CIFAR10 | C10/Mix | | | | |
|---|---|---|---|---|---|---|
| Ratio $P$ | 0% | 50% | 70% | 80% | 90% | 95% |
| ER | 20.30$\pm$4.50 | 20.20$\pm$3.16 | 22.77$\pm$4.62 | 22.05$\pm$5.46 | 25.60$\pm$3.91 | 26.89$\pm$4.03 |
| DER++ | 17.36$\pm$2.62 | 17.76$\pm$2.03 | 19.10$\pm$3.82 | 18.49$\pm$3.86 | 21.71$\pm$7.22 | 20.94$\pm$4.21 |
| ERACE | 9.59$\pm$2.12 | 10.67$\pm$2.66 | 11.61$\pm$2.12 | 13.58$\pm$4.21 | 14.09$\pm$4.70 | 12.06$\pm$2.61 |
| OCM | 10.43$\pm$2.51 | 11.04$\pm$2.82 | 12.89$\pm$3.58 | 14.24$\pm$3.79 | 15.01$\pm$2.95 | 15.15$\pm$2.33 |
| GSA | 16.25$\pm$2.62 | 15.33$\pm$4.76 | 16.34$\pm$3.71 | 16.28$\pm$3.63 | 15.94$\pm$2.53 | 16.46$\pm$2.97 |
| OnPro | 12.07$\pm$3.19 | 12.72$\pm$2.70 | 12.93$\pm$3.08 | 13.39$\pm$3.11 | 14.36$\pm$2.92 | 14.49$\pm$3.65 |
| ESRM | 26.12$\pm$2.05 | 23.75$\pm$2.39 | 21.81$\pm$2.53 | 21.83$\pm$2.24 | 22.77$\pm$1.15 | 23.48$\pm$2.78 |

| Dataset | CIFAR100 | C100/Mix | | | | |
|---|---|---|---|---|---|---|
| Ratio $P$ | 0% | 50% | 70% | 80% | 90% | 95% |
| ER | 23.86$\pm$2.73 | 25.47$\pm$3.32 | 25.92$\pm$3.13 | 25.18$\pm$2.30 | 26.70$\pm$2.79 | 27.95$\pm$3.58 |
| DER++ | 33.11$\pm$4.58 | 34.53$\pm$3.63 | 36.41$\pm$3.53 | 36.96$\pm$4.55 | 37.83$\pm$2.08 | 39.97$\pm$2.38 |
| ERACE | 9.81$\pm$1.72 | 11.32$\pm$1.66 | 14.62$\pm$3.75 | 15.33$\pm$3.12 | 16.16$\pm$3.43 | 18.34$\pm$3.36 |
| OCM | 8.35$\pm$2.07 | 9.87$\pm$2.31 | 9.98$\pm$2.21 | 11.12$\pm$3.01 | 13.87$\pm$3.72 | 14.54$\pm$3.62 |
| GSA | 19.62$\pm$2.73 | 20.00$\pm$1.83 | 20.71$\pm$1.27 | 22.26$\pm$2.97 | 22.06$\pm$3.03 | 22.01$\pm$4.44 |
| OnPro | 11.91$\pm$1.87 | 14.28$\pm$2.69 | 13.79$\pm$1.32 | 14.50$\pm$2.16 | 15.71$\pm$3.13 | 16.36$\pm$4.01 |
| ESRM | 32.67$\pm$1.46 | 32.72$\pm$1.77 | 31.34$\pm$1.26 | 30.19$\pm$2.04 | 29.94$\pm$1.34 | 30.19$\pm$1.70 |

| Dataset | Tiny | Tiny/Mix | | | | |
|---|---|---|---|---|---|---|
| Ratio $P$ | 0% | 50% | 70% | 80% | 90% | 95% |
| ER | 61.84$\pm$2.65 | 69.86$\pm$3.33 | 74.38$\pm$2.97 | 77.56$\pm$3.09 | 81.40$\pm$3.07 | 87.49$\pm$3.14 |
| DER++ | 72.51$\pm$5.53 | 78.64$\pm$4.06 | 83.17$\pm$1.41 | 87.87$\pm$3.32 | 93.01$\pm$1.55 | 96.49$\pm$1.26 |
| ERACE | 36.40$\pm$2.74 | 43.29$\pm$2.72 | 48.53$\pm$3.15 | 55.66$\pm$3.50 | 66.81$\pm$3.03 | 74.77$\pm$3.71 |
| OCM | 32.25$\pm$1.44 | 38.31$\pm$2.25 | 45.04$\pm$3.00 | 49.95$\pm$2.56 | 61.10$\pm$1.94 | 74.39$\pm$4.50 |
| GSA | 44.78$\pm$2.76 | 51.89$\pm$4.09 | 57.09$\pm$5.29 | 63.64$\pm$4.54 | 71.41$\pm$3.71 | 81.18$\pm$4.13 |
| OnPro | 42.81$\pm$4.63 | 52.02$\pm$3.70 | 58.82$\pm$3.49 | 63.49$\pm$4.40 | 76.68$\pm$3.02 | 82.61$\pm$6.06 |
| ESRM | 61.53$\pm$1.36 | 64.21$\pm$1.54 | 62.78$\pm$1.66 | 62.82$\pm$2.15 | 67.37$\pm$2.25 | 71.86$\pm$2.36 |

Table 10: Relative Forgetting (%; lower is better) on contaminated datasets with various contamination ratio $P$.

| Dataset | C100(M=1k) | C100/SDXL (M=1k) | | | | |
|---|---|---|---|---|---|---|
| Ratio $P$ | 0% | 50% | 70% | 80% | 90% | 95% |
| ER | $24.92_{\pm1.33}$ | $23.25_{\pm0.98}(-1.67)$ | $22.18_{\pm1.58}(-2.74)$ | $20.63_{\pm1.27}(-4.29)$ | $18.96_{\pm1.54}(-5.96)$ | $17.79_{\pm0.97}(-7.13)$ |
| DER++ | $25.86_{\pm2.43}$ | $21.92_{\pm1.36}(-3.94)$ | $19.47_{\pm1.14}(-6.39)$ | $16.69_{\pm1.65}(-9.17)$ | $13.60_{\pm1.06}(-12.26)$ | $11.15_{\pm1.31}(-14.71)$ |
| ERACE | $\mathbf{28.22_{\pm1.09}}$ | $23.96_{\pm0.92}(-4.26)$ | $19.38_{\pm0.78}(-8.84)$ | $17.71_{\pm0.90}(-10.51)$ | $13.85_{\pm0.71}(-14.37)$ | $11.94_{\pm0.43}(-16.28)$ |
| OCM | $28.02_{\pm0.74}$ | $26.54_{\pm1.02}(-1.48)$ | $24.94_{\pm1.07}(-3.08)$ | $24.17_{\pm1.03}(-3.85)$ | $22.12_{\pm0.64}(-5.90)$ | $20.65_{\pm1.08}(-7.37)$ |
| GSA | $28.15_{\pm1.59}$ | $26.20_{\pm1.47}(-1.95)$ | $23.52_{\pm1.14}(-4.63)$ | $22.90_{\pm0.91}(-5.25)$ | $19.20_{\pm1.27}(-8.95)$ | $16.89_{\pm1.37}(-11.26)$ |
| OnPro | $26.92_{\pm1.31}$ | $26.31_{\pm1.30}(-0.61)$ | $24.85_{\pm0.94}(-2.07)$ | $23.10_{\pm1.34}(-3.82)$ | $20.98_{\pm1.02}(-5.94)$ | $19.77_{\pm1.49}(-7.15)$ |
| ESRM | $27.14_{\pm0.79}$ | $\mathbf{26.57_{\pm1.10}(-0.57)}$ | $\mathbf{26.17_{\pm0.91}(-0.97)}$ | $\mathbf{25.05_{\pm0.50}(-2.09)}$ | $\mathbf{24.43_{\pm0.91}(-2.71)}$ | $\mathbf{23.35_{\pm1.17}(-3.79)}$ |

| Dataset | C100(M=2k) | C100/SDXL (M=2K) | | | | |
|---|---|---|---|---|---|---|
| Ratio $P$ | 0% | 50% | 70% | 80% | 90% | 95% |
| ER | $32.07_{\pm1.51}$ | $30.24_{\pm1.15}(-1.83)$ | $27.38_{\pm1.41}(-4.69)$ | $25.95_{\pm1.22}(-6.12)$ | $23.68_{\pm1.14}(-8.39)$ | $21.79_{\pm0.65}(-10.28)$ |
| DER++ | $33.37_{\pm2.11}$ | $27.95_{\pm2.12}(-5.42)$ | $24.60_{\pm1.53}(-8.77)$ | $21.07_{\pm1.89}(-12.30)$ | $17.39_{\pm1.06}(-15.98)$ | $14.15_{\pm1.52}(-19.22)$ |
| ERACE | $34.30_{\pm1.49}$ | $28.69_{\pm1.71}(-5.61)$ | $24.01_{\pm1.14}(-10.29)$ | $21.23_{\pm0.84}(-13.07)$ | $16.47_{\pm0.95}(-17.83)$ | $13.25_{\pm1.68}(-21.05)$ |
| OCM | $35.69_{\pm1.36}$ | $32.39_{\pm1.09}(-3.30)$ | $31.15_{\pm0.84}(-4.54)$ | $28.38_{\pm1.28}(-7.31)$ | $26.76_{\pm0.79}(-8.93)$ | $24.75_{\pm0.67}(-10.94)$ |
| GSA | $35.31_{\pm1.47}$ | $32.72_{\pm1.33}(-2.59)$ | $28.97_{\pm1.23}(-6.34)$ | $26.87_{\pm1.03}(-8.44)$ | $23.42_{\pm1.17}(-11.89)$ | $19.80_{\pm1.32}(-15.51)$ |
| OnPro | $33.52_{\pm0.80}$ | $31.33_{\pm0.75}(-2.19)$ | $30.02_{\pm1.01}(-3.50)$ | $27.90_{\pm0.85}(-5.62)$ | $24.38_{\pm0.69}(-9.14)$ | $22.58_{\pm1.18}(-10.94)$ |
| ESRM | $\mathbf{36.25_{\pm0.79}}$ | $\mathbf{34.55_{\pm1.33}(-1.70)}$ | $\mathbf{34.13_{\pm1.03}(-2.12)}$ | $\mathbf{33.77_{\pm1.08}(-2.48)}$ | $\mathbf{31.70_{\pm0.86}(-4.55)}$ | $\mathbf{29.21_{\pm0.91}(-7.04)}$ |

Table 11: Final Average Accuracy (%; higher is better) on C100/SDXL dataset with different memory size $M$ and contamination ratio $P$. Numbers in parentheses indicate the performance degradation due to contamination compared to the clean setting. The average and deviation over 10 runs are reported.

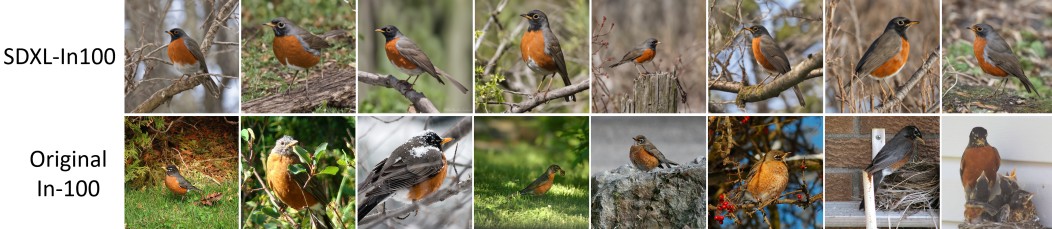

Figure 10: Random sampled images from class "n01558993" (Robin) in SDXL-In100 and original ImageNet-100 dataset. For clarity, we have cropped some backgrounds and resized the vanilla ImageNet-100 samples.

# D  Implementation Details

## D.1  Dataset.

As discussed in Sec. 3, we used four benchmark datasets in evaluation, including CIFAR-10/100, TinyImageNet, and ImageNet-100. In the experiments, all of the datasets are split into tasks containing non-overlapping classes. The details about the task split are as follows:

**CIFAR-10** [24] has ten classes with 50,000 training images and 10,000 test images. The image is $32 \times 32$ in size. The dataset is split into five disjoint tasks with two classes per task.

**CIFAR-100** [24] has 100 classes with 50,000 training samples and 10,000 test samples. Image size is $32 \times 32$. It is split into 10 disjoint tasks with 10 classes per task.

**TinyImageNet** [25] has 200 classes, 100,000 training samples, and 10,000 test samples. Image size is $64 \times 64$. The dataset is split into 100 non-overlapping tasks with two classes per task.

**ImageNet-100** [20] is a subset of ImageNet-1k [13] dataset. It consists of 100 classes. We follow [11] for the class selection. We do not perform image resizing in generating ImageNet-100 from the original ImageNet-1k dataset. The dataset is split into 10 disjoint classes with 10 classes per task.

## D.2  Details about synthetic dataset generation.

**Image size of SDXL-IN100.** In the synthetic dataset generation, we manually adjust the size of generated images to match that of the original dataset. For the ImageNet-100 dataset, since the image size is not fixed, we resize the generated images to $224 \times 224$, to align with the training protocols.

| Generative Model | Diffusion Steps | Upsample(Refiner) Steps | Guidance Scale |
|:---:|:---:|:---:|:---:|
| SD1.4 | 50 | N/A | 7.5 |
| SD2.1 | 50 | N/A | 7.5 |
| SDXL | 40 | 40 | 5.0 |
| VQDM | 100 | N/A | 7.5 |
| GLIDE | 100 | 27 | 3.0 |

Table 12: Hyperparameters used in image generation.

**Class-wise distribution of sources for Mix-C10/C100/Tiny dataset.** In the main manuscript, we mentioned that the dataset of setting **b)** in Sec. 3.1 is generated from five synthetic models: Stable Diffusion v1.4, Stable Diffusion v2.1, Stable Diffusion XL, VQDM, and GLIDE. Each method contributes 20% of the dataset in setting **b)**. In our implementation, we ensure that this distribution is consistent across all classes in the dataset, so that each class has an equal number of images from each generation model.

**Hyperparameters used in image generation.** For Stable Diffusion and VQDM, we use source code and model snapshots from huggingface, as mentioned in Table 14. For Glide experiments, we use the official implementation and the released model snapshots. Following the recommendation, we use the refiner in Stable Diffusion XL and the upsampler in GLIDE. The diffusion steps and guidance scale hyperparameters we used are shown in Table 12. For other hyperparameters, we follow the recommendations from Huggingface and GLIDE's official implementation. We use the prompt "An image of a class_name." as the text guidance to generate the image and interpolate the generated image to the size of the target dataset (32 for CIFAR, 64 for TinyImageNet, and 224 for ImageNet-100).

**Samples from the generated datasets.** Fig. 10 shows some samples from class "n01558993" (Robin) in the SDXL-In100 dataset along with the samples from the original dataset. We can notice a significant loss of diversity in the samples from SDXL-In100.

### D.3  Details about synthetic contamination simulation.

As mentioned in Sec. 3.2, we generate synthetic twins of benchmark datasets and substitute a fixed portion $P$ of the original datasets with their synthetic counterparts. Similar to the class-wise distribution of Mix-C10/C100/Tiny, we also conduct the mixture class-wise. For datasets in setting **a)**, we assure that the contamination ratio in each class is also $P$. For datasets in setting **b)**, while maintaining a consistent class-wise contamination ratio, we also ensure that each individual synthetic model contributes 20% of the contamination in each class.

### D.4  Task sequence.

In some work, the authors use a fixed task sequence for fair comparison. However, the final performance is largely affected by the task order. For fair comparison, we randomly assign the class to tasks and shuffle the sequence of tasks with 10 fixed random seeds. This ensures the evaluation is not biased to task difficulty.

### D.5  Data augmentation.

Data augmentation is effective in boosting the training of online continual learners. Methods may benefit differently from different augmentation intensities, and some methods may favor simpler augmentations instead of complicated ones. For a fair comparison, it is vital to ensure all methods are in their optimal performance. Thus, we introduce two different augmentation strategies:

**1. Partial strategy.** The partial augmentation is a weaker version of augmentation, consisting of random cropping with $p = 0.5$, followed by random horizontal flip with $p = 0.5$.

**2. Full strategy.** The full augmentation strategy is a stronger version of augmentation. The full augmentation strategy is a superset of its partial counterpart, which consists of random cropping, random horizontal flip, color jitter, and random grayscale. The parameters for color jitter are set to $(0.4, 0.4, 0.4, 0.1)$ with $p = 0.8$. The probability of random grayscale is set to $p = 0.2$.

We define the data augmentation strategy of each method with a hyperparameter search, as detailed in Appendix D.6.

## D.6 Hyperparameter search protocol.

For hyperparameters in all of the methods (except DER++ on TinyImageNet), we conduct a hyperparameter search on the clean CIFAR-100 dataset with a memory size of 5,000, and apply the same hyperparameter to all of the other settings. The exhaustive list of the hyperparameter search is shown in Table 13.

**Special treatment for DER++.** DER++ encounters a catastrophic performance defect (close to 0) when trained on the TinyImageNet dataset using an optimizer with momentum. Thus, we applied another hyperparameter search for DER++ on TinyImageNet and found the SGD optimizer (without Momentum) gives reasonable performance. All the experiments of DER++ on TinyImageNet are conducted using these new hyperparameters.

## D.7 Hardware and computation.

All of the experiments are conducted on NVIDIA A100 GPUs. The average training time of each method on CIFAR-100 (Memory size = 5k), ImageNet-100 (Memory size = 5k), and TinyImageNet (Memory size = 10k) is shown in Fig. 11. The training efficiency is much faster than OCM and OnPro, while on par with the most efficient method.

## D.8 Useful source code links.

For continual learning baselines, we use the codebase listed in Table 14 to reimplement baseline methods. For image generation methods, we use the Diffuser library from Hugging Face for Stable Diffusion and VQDM experiments, and we use the codebase in Table 14 for GLIDE.

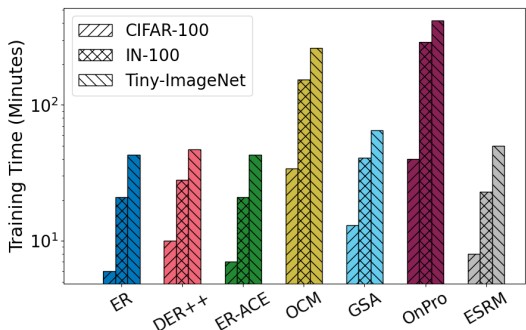

Figure 11: The average training time of each method trained on CIFAR-100 (M=5k), IN-100 (M=5k), and TinyImageNet (M=10k) dataset. For better readability, the values are plotted on the logarithm scale. The numbers are averaged from 10 runs.

| Method | HP | Values |
|---|---|---|
| ER | optimizer | [SGD, AdamW] |
| | lr | [0.1, 0.05, 0.01, 0.005, 0.001, 0.0005, 0.0001] |
| | weight decay | [0, 1e-4] |
| | momentum (for SGD) | [0, 0.9] |
| | aug. strat. | [partial, full] |
| DER++ | optimizer | [SGD, AdamW] |
| | lr | [0.1, 0.05, 0.01, 0.005, 0.001, 0.0005, 0.0001] |
| | weight decay | [0, 1e-4] |
| | momentum (for SGD) | [0, 0.9] |
| | aug. strat. | [partial, full] |
| | alpha | [0.1, 0.2, 0.5, 1.0] |
| | beta | [0.5, 1.0] |
| ER-ACE | optimizer | [SGD, AdamW] |
| | lr | [0.1, 0.05, 0.01, 0.005, 0.001, 0.0005, 0.0001] |
| | weight decay | [0, 1e-4] |
| | momentum (for SGD) | [0, 0.9] |
| | aug. strat. | [partial, full] |
| OCM | optimizer | [AdamW] |
| | lr | [0.001] |
| | weight decay | [1e-4] |
| | aug. strat. | [partial, full] |
| GSA | optimizer | [SGD, AdamW] |
| | lr | [0.1, 0.05, 0.01, 0.005, 0.001, 0.0005, 0.0001] |
| | weight decay | [0, 1e-4] |
| | momentum (for SGD) | [0, 0.9] |
| | aug. strat. | [partial, full] |
| OnPro | optimizer | [SGD, AdamW] |
| | lr | [0.1, 0.05, 0.01, 0.005, 0.001, 0.0005, 0.0001] |
| | weight decay | [0, 1e-4] |
| | momentum (for SGD) | [0, 0.9] |
| | aug. strat. | [partial, full] |
| ESRM | optimizer | [SGD, AdamW] |
| | lr | [0.1, 0.05, 0.01, 0.005, 0.001, 0.0005, 0.0001] |
| | weight decay | [0, 1e-4] |
| | momentum (for SGD) | [0, 0.9] |
| | aug. strat. | [partial, full] |
| | $\lambda_1$ | [0.1, 0.2, 0.5, 1, 2, 5] |
| | $\lambda_2$ | [0.1, 0.2, 0.5, 1, 2, 5] |

Table 13: Exhaustive list of hyperparameters searched on CIFAR-100.

| Baseline | Links |
|---|---|
| ER & ER-ACE | https://github.com/pclucas14/AML |
| DER++ | https://github.com/aimagelab/mammoth |
| OCM | https://github.com/gydpku/OCM |
| GSA | https://github.com/gydpku/GSA |
| OnPro | https://github.com/weillllllls/OnPro |
| Stable Diffusion & VQDM | https://github.com/huggingface/diffusers |
| GLIDE | https://github.com/openai/glide-text2im |

Table 14: Baselines and their source code URLs.

