# OpenReview forum: "Dealing with Synthetic Data Contamination in Online Continual Learning"
_NeurIPS.cc/2024/Conference — NeurIPS 2024 poster_

### Official Review · Reviewer_h2ny · 2024-06-28

**Soundness:** 3
**Presentation:** 3
**Contribution:** 3
**Rating:** 7
**Confidence:** 3

**Summary:**

This paper  investigates the impact of AI-generated images on the performance of online continual learning (CL) models. It introduces a novel method called Entropy Selection with Real-synthetic similarity Maximization (ESRM) to mitigate the negative effects of synthetic data contamination. ESRM leverages entropy-based sample selection and a contrastive learning approach to align the feature embeddings of real and synthetic data, thereby enhancing the robustness of CL models against the degradation caused by synthetic data.

**Strengths:**

- The authors clearly articulate the problem, methodology, and results, enhancing the paper's accessibility and understanding for a broad readership.

- The paper uniquely identifies the issue of synthetic data contamination in online continual learning, a significant challenge for the future of this field. The work has substantial implications for the ML community, offering a pioneering approach to maintaining the integrity of continual learning models in the presence of synthetic data.

- This paper proposes ESRM, an innovative method that combines entropy selection and contrastive learning to mitigate the negative effects of synthetic data, demonstrating creativity in addressing this new problem.

- The paper is underpinned by robust technical approaches, ensuring the quality and reliability of the proposed solution through comprehensive experimental validation.

**Weaknesses:**

- The paper evaluates the impact of synthetic data using a limited set of generative models. And the experiments are primarily conducted on image classification datasets.  Expanding this to include a broader range of tasks, particularly truly text-to-images, text-to-text, could strengthen the findings.

- The method for generating synthetic datasets is straightforward, using simple prompts. Incorporating more complex and diverse prompts, potentially using large language models to simulate user queries, could better reflect real-world synthetic data contamination.

- The reliance on entropy as a key metric for distinguishing real from synthetic data might not be universally applicable. Further exploration of this metric's effectiveness across different domains and its theoretical underpinnings could bolster the method's credibility.

- While the paper provides insights into the method's effectiveness, there is limited discussion on its computational efficiency and scalability, especially when dealing with large-scale datasets or high contamination ratios. Addressing these aspects could be crucial for practical applications where computational resources and time are critical constraints.

**Questions:**

Please refer to the weaknesses above.

**Limitations:**

The authors have discussed the limitations in the paper.

---

> ### Author Rebuttal · Authors · 2024-08-07
>
> Thank you for your feedback and valuable comments.
>
> ## Weaknesses
> 1. We indeed tested our method with a limited set of generative models, due to the computation constraint. Also, it is true that almost all of the existing work in online Continual Learning (CL) is limited to Class-Incremental Learning (CIL) scenarios, where the major task is image classification. We fully agree the scope of the tasks can be beneficial, which we take as an important direction for future work.
>
> 2. Following the suggestion, we conducted extra rebuttal experiments as shown in the general rebuttal (1). We use LLaMA to generate more diverse and complex prompts to simulate a more realistic contamination scenario. The experimental result in Table I in the rebuttal material validates the effectiveness of our method when the synthetic images are generated with LLM-enhanced prompts.
>
> 3. The entropy criteria is indeed important in our work, and we would like to prove the universality of the entropy metric both empirically and theoretically.
>    Empirically, we expand the experiments with broader settings. The Domain-Incremental Learning setting and LLM enhancement setting in Table I of the rebuttal material further proved the universality of ESRM.
>    Theoretically, classifiers trained on a limited diversity dataset often overfit, resulting in confident predictions on the training data but poor generalization to new data. Such a pattern is universal and not limited to synthetic contamination. Methods like Label Smoothing [4] and Confidence Penalty [5] are proposed to alleviate such problems. In our paper, this limited diversity issue is magnified by synthetic contamination, where the generative model's output diversity remains imperfect for current AI research. With limited diversity, synthetic data are easier for the model's feature extractor to cluster (cf. Fig. 3 in the paper) and for the classifier to classify, leading to more confident (lower entropy) predictions (cf. Fig. 2 in our paper). We will include such experiments and theoretical discussions to further improve our manuscript.
>
> 4. We regard computation efficiency and scalability as crucial criteria for our method, especially in online continual learning scenarios. Thus, we included the training time on the CIFAR-100 dataset in Sec. D.7 and Fig. 11 in the appendix. Following the suggestion, to show how the computation of different baselines is scaled with larger datasets, in general rebuttal (4), we include the training time of all baselines on TinyImageNet and ImageNet-100 datasets. As shown in Fig. II of the rebuttal material, the training time of our method is significantly faster than some state-of-the-art methods like OCM and OnPro, while it is on par with the most efficient method. Also, we would like to clarify that the training computation is agnostic with contamination ratio, because in the generation of the contaminated dataset, we *replace* the real images with synthetic images while not changing the size of the dataset.
>
> Reference:
>
> [4] "Rethinking the inception architecture for computer vision." CVPR 2016
>
> [5] "Regularizing neural networks by penalizing confident output distributions." arXiv:1701.06548 (2017)

---

> > ### Comment · Reviewer_h2ny · 2024-08-12
> > **Thank you for your rebuttal**
> >
> > Thank you for the rebuttal.  This paper is very insteresting and promising. After considering the comments from other reviewers and the rebuttal, I decided to raise my score.

---

> > > ### Author Response · Authors · 2024-08-12
> > >
> > > Thank you for your feedback and valuable comments.

---

### Official Review · Reviewer_pFAa · 2024-07-06

**Soundness:** 3
**Presentation:** 4
**Contribution:** 3
**Rating:** 7
**Confidence:** 3

**Summary:**

Image generation has been showing promising achievements in the last few years, but the generative models may not be able to keep up with the distribution of the real samples. Due to low diversity, synthetic images perform poorly in downstream learning tasks. This paper tackles this problem by diversifying the sample selection using the proposed ESRM framework, which prioritizes the high-entropy samples in the experience replay memory with an RM loss function. The authors first performed experimental analysis to build their motivation in section 4 and proposed their solution to the problem in section 5. The experiments are diversified with several datasets, benchmarks, and extensive ablation studies.

**Strengths:**

1. The paper is nicely written, with clear language and sufficient details.

3. Section 4's experimental analysis of existing CL methods clearly motivates the proposed model.

4. The authors propose a novel loss function to address the issue of low-entropy samples in CL. This approach builds upon existing supervised constructive loss techniques. Their method aims to maximize the cosine similarity between embeddings of real and synthetic samples while simultaneously minimizing the impact of low-entropy samples on the learning process.

4. In the ablation study, the authors studied different components of the proposed framework that show the superiority of the overall ESRM.

**Weaknesses:**

1. The paper only deals with diffusion generative models. I understand that diffusion models are the most powerful image generators but it would be good to experiment other generative models.

2. The accuracy scores are overall low in CL. Does the proposed method

3. Why is a fixed 50% dropping rate selected for ES?

4. Downstream ML tasks have been explored with synthetic data to validate synthetic datasets, and related works clearly miss that angle. The authors should also discuss them and maybe add some of them to benchmarks.

**Questions:**

Authors can refer to the weaknesses section for major questions. Here, I listed minor points for authors reference:

1. It seems there is a typo in the line 231.
2. It could be good to highlight the best results in Tables 3 and 4, the same as Table 2.

**Limitations:**

I agree with the limitation the authors brought up.

---

> ### Author Rebuttal · Authors · 2024-08-07
>
> Thank you for your feedback and valuable comments.
>
> ## Weaknesses
>
> 1. In our experiments, we prioritize the importance of diffusion-based contamination. There are two reasons: Firstly, training with synthetic images generated with GAN-based or autoencoder-based methods yields catastrophic performance, especially when the contamination is severe. Another reason is that we find GAN-based or autoencoder-based synthetic images are relatively easy to distinguish from their real counterparts, which hinders them from constituting a de facto “contamination”. For example, in our preliminary experiments, we tried to use GALIP [1], a state-of-the-art text-to-image GAN-based generative method to generate images with our text prompt, as shown in the general rebuttal (6). Fig. III in the rebuttal material shows the GALIP is subject to an extremely limited diversity problem. While we prioritize diffusion-based contamination, we regard using LLM-enhanced prompts to diversify GALIP-generated images as an important future direction.
>
> 2. In this work, we focus on the online Continual Learning (CL) setting specifically, where the synthetic contamination is a more factual and severe issue because the online setting restricts us from assessing the quality of the training data beforehand. Unfortunately, due to the online restrictions, there is still a salient performance gap between the performance of conventional CL and the state-of-the-art online CL. Also, it is noteworthy that ESRM could achieve a clean performance that is on par with the current state-of-the-art methods (OCM, GSA, and OnPro). We hope our research sheds light on the research field and helps narrow down the gap.
>
> 3. Intuitively and factually, the dropping rate in ES is a hyperparameter that is affected by the contamination ratio of the training dataset. There are two reasons to choose a fixed (50%) dropping rate. **(a)** We assume the continual learner does not have prior information about the contamination ratio of the dataset, which is more realistic in the real world. This is also the reason why we do not and should not perform a hyperparameter search for the dropping rate for each contamination ratio. **(b)** A drop rate of 50% gives competitive performance even under extreme conditions (Contamination ratio = 95%). We believe that the obtained performances demonstrate that a fixed ES dropping rate is resilient to different conditions (dataset, contamination ratio).
>
> 4. There is indeed excellent work on validating synthetic data, for example, UniFD [2], FatFormer [3], etc. We will include such discussions in the related work section in the revised manuscript. Experimentally, we replaced the ESRM's entropy selection strategy with the pretrained UniFD synthetic data detector, as illustrated in the general rebuttal (5). With our test, the accuracy of the UniFD detector is 66.19% on the C100/SDXL dataset. Table II in the rebuttal material shows that UniFD is more effective than random selection, but its performance is still limited. This is due to a distribution mismatch between our dataset and the dataset used to train the detectors. Moreover, due to the online constraint, we had to set the threshold of UniFD at 0.5, because we can not search the optimal parameter without the knowledge of the dataset distribution.
>
> ## Questions
>
> &emsp; Thank you for your suggestions, we will carefully revise the manuscript and update the tables for a better readability of our paper.
>
> Reference:
>
> [1] "GALIP: Generative adversarial clips for text-to-image synthesis." CVPR 2023
>
> [2] "Towards universal fake image detectors that generalize across generative models." CVPR 2023
>
> [3] "Forgery-aware adaptive transformer for generalizable synthetic image detection." CVPR 2024

---

### Official Review · Reviewer_ttuB · 2024-07-15

**Soundness:** 3
**Presentation:** 3
**Contribution:** 2
**Rating:** 5
**Confidence:** 3

**Summary:**

This paper investigates the negative impact of synthetic data contamination on existing online continual learning methods. An entropy selection with the real-synthetic similarity maximization method is proposed to alleviate the performance deterioration.

**Strengths:**

1. Detailed analysis of synthetic data contamination and its influence on continual learning.
2. This paper is technically clear and easy to follow.

**Weaknesses:**

1. The creation of simulated data is the cornerstone of this research, but there is a lack of detailed explanation on how to generate these data in the main paper.
2. For Observation 4 "With the limited diversity of synthetic data", how about mixing the synthetic data from different generation models to increase the diversity, since the synthetic images on the internet also form different models? While the observation remains unchanged in this case?
3. In order to simulate real data collection more realistically, in addition to synthetic data, new open-domain real data should also be incorporated.

**Questions:**

Please refer to the weaknesses.

**Limitations:**

The limitations are discussed while the potential negative societal impact is not discussed. However, for this work, I think it is not necessary to discuss this.

---

> ### Author Rebuttal · Authors · 2024-08-07
>
> Thank you for your feedback and valuable comments.
>
> ## Weaknesses
>
> 1. We would like to include more detailed information on the synthetic dataset generation process. For Stable Diffusion and VQDM, we use source code and model snapshots from huggingface, as mentioned in Table 12 in the appendix. For Glide experiments, we use the official implementation and the released model snapshots. Following the recommendation, we use the refiner in Stable Diffusion XL and the upsampler in GLIDE. The diffusion steps and guidance scale hyperparameters we use are as follows:
>
>     | Generative Model | Diffusion Steps | Upsample(Refiner) Steps | Guidance Scale |
>     |:-----------------:|:-----------------:|:----------------:|:----------------:|
>     | SD1.4            | 50              | N/A            | 7.5            |
>     | SD2.1            | 50              | N/A            | 7.5            |
>     | SDXL             | 40              | 40             | 5.0            |
>     | VQDM             | 100             | N/A            | 7.5            |
>     | GLIDE            | 100             | 27             | 3.0            |
>
>     For other hyperparameters, we follow the recommendations from Huggingface and GLIDE's official implementation. We use the prompt "An image of a {class_name}." as the text guidance to generate the image and interpolate the generated image to the size of the target dataset (32 for CIFAR, 64 for TinyImageNet, and 224 for ImageNet-100). We will include such detailed information in the revised manuscript to help the reproducibility of our paper. Also, for reproducibility, we will include the source code for synthetic dataset generation in our project codebase and make it publicly available.
>
> 2. As suggested in Sec. 3.1, we have two different strategies for simulating synthetic data contamination: **(a)** using data generated from SDXL only and **(b)** mixing the data generated from different generative models. In Table 3 of our paper, We have included the t-SNE visualization result on **(a)** setting to ground the observation 4. For **(b)** setting, besides the final average accuracy shown in Table 7 in the appendix, we include extra t-SNE visualization results in the rebuttal material, as introduced in the general rebuttal (3). The t-SNE visualization of feature representations given in Fig. I(a) of the rebuttal material shows that even when generating synthetic data with a mixture of generation models, a gap still exists in feature space between generated and non-generated data when training with ER. Such experiments additionally support the claims of Observation 4.
>
>
> 3. Following the suggestion, we extended our experiments from Class-Incremental Learning (CIL) only to include Domain-Incremental Learning (DIL) experiments on the DIL-CIFAR20 dataset. As mentioned in the general rebuttal (2), the DIL results of ESRM outperform other baselines, showing the robustness of ESRM against synthetic contamination.

---

> > ### Comment · Reviewer_ttuB · 2024-08-12
> > **Thanks for the  rebuttal.**
> >
> > The authors have addressed most of my concerns. I decide to keep my initial positive rating.

---

> > > ### Author Response · Authors · 2024-08-12
> > >
> > > Thank you for your feedback and insightful comments.

---

### Author Rebuttal · Authors · 2024-08-07

We would like to thank the reviewers for their comments and suggestions, which enabled us to improve the manuscript significantly. We respond to each reviewer individually. Here, we introduce the rebuttal experiments in the attached PDF file.

## 1. Results with LLM Enhanced Prompts
As suggested by reviewer h2ny, we expand the synthetic dataset generation from the baseline prompt ("An image of {class_name}") to the LLM-enhanced prompt. We leverage the open-source instruction-fine-tuned LLaMA-3 model with 8 billion parameters (`meta-llama/Meta-Llama-3-8B-Instruct`) to generate the enhanced prompt for CIFAR-100 dataset. For each class, 50 prompts are generated, and we use Stable Diffusion XL to generate 10 images per prompt to formulate the LLM-enhanced CIFAR-100 dataset (denoted as C100/SDXL-LLaMA). Some examples of LLM-generated prompts are as follows:
1. A juicy red oval apple with a tiny worm hole is sitting on the wooden kitchen counter.
2. A vibrant orange tropical aquarium fish with iridescent scales swims lazily in a large glass tank on the sunlit windowsill.

As shown in Table I in the rebuttal material, for all baseline methods, when the synthetic dataset is generated with LLM-enhanced prompts, the performance deterioration of synthetic contamination is still significant. Even with such advanced prompts, ESRM can significantly alleviate the performance deterioration while achieving satisfactory results. Notably, LLM-enhanced results (C100/SDXL-LLaMA in Table I of the rebuttal material) sometimes bring performance drops compared with non-LLM-enhanced results (C100/SDXL in Table 2 of our paper). This is because the LLM might introduce undesired items into the language prompt.

## 2. Domain-Incremental Learning (DIL) results
While our current experiment mainly focuses on the Class-Incremental Learning (CIL) setting of online Continual Learning, we include extra experiments in Domain-Incremental Learning (DIL) scenarios, following the suggestion from reviewers ttuB and h2ny.
We conducted the experiment with the 20 coarse labels of the CIFAR-100 dataset. Since the 100 classes in CIFAR-100 are grouped into 20 superclasses with 5 fine-grained classes for each superclass, we split the CIFAR-100 dataset with 5 domain increment steps. For each step, we feed the model with the training data of a fine-grained class for each superclass. We refer to this dataset as DIL-CIFAR20, as the model only classifies coarse labels.

Similar to the simulated CIFAR100/SDXL dataset, we replace the images in the DIL-CIFAR20 dataset with its Stable Diffusion XL generated counterpart with a contamination ratio P, as per the protocol in Sec. 3.2.

Table I of the rebuttal material shows the final average accuracy with different contamination ratios. Notably, we adapted the CIL-specifically designed components in OnPro and GSA to the DIL scenario, and the performance suffered a decent loss. We did not report ERACE results because its Asymmetric Cross Entropy (ACE) loss converges to standard cross-entropy loss in the DIL scenario, making it equivalent to vanilla ER.

The experimental results show that ESRM can yield robust performance against domain shift in the DIL setting, under different synthetic contamination situations, which validates the efficiency of ESRM under DIL settings.

## 3. Visualization on C100/Mix dataset
As suggested by reviewer ttuB, to better ground the claim in observation 4, we included extra visualization experiments on the C100/Mix dataset, where the synthetic data are generated from the mixture of different generative models. From the t-SNE visualization result in Fig. I of the rebuttal material, we can see that similar to Fig. 3 in the paper, the feature gap of the baseline method (ER) in the embedding space still holds even when the synthetic data is generated with different generative models. Also, with ESRM, the feature misalignment problem is alleviated.

## 4. Training time on other datasets
We have shown the training time on the CIFAR-100 dataset in Fig. 11 in the appendix. As suggested by reviewer h2ny, to show how different methods scale larger datasets, we include the training time of all baselines on TinyImageNet and ImageNet-100 datasets. As shown in Fig. II in the attached material, we plot the training time on the logarithm scale for better readability. The computation efficiency of ESRM is always on par with the most efficient method (ER), while significantly outperforming OCM and OnPro.

## 5. Benchmarking Pre-trained synthetic image detectors
As suggested by reviewer pFAa, we replace ES in our memory strategy with UniFD [2], a pre-trained synthetic image detector. We use the prediction of UniFD to validate the synthetic status (i.e., whether the image is real or synthetic) of the training images, and store the real images in the memory buffer. As shown in Table II of the rebuttal material, UniFD outperforms random selection but still has limited effectiveness despite leveraging extra-information from pre-training. One explanation is that it's not designed for online learning scenarios. Due to the online constraint, we set the threshold of UniFD to 0.5, instead of searching for the best threshold parameter (because we do not know the dataset distribution). Also, the domain gap between the UNiFD training set and our dataset further limits the performance.

## 6. Image Generation Results with GANs
We include the image generation result with GALIP [1], a state-of-the-art text-to-image GAN model. Following our image generation protocol, we use the prompt "An image of {class_name}" to generate synthetic images. We use the GALIP model pretrained on CC12M dataset. As shown in Fig. III in the attached material, the generation result of GALIP suffers from a huge lack of diversity problem.

Reference:

[1] "GALIP: Generative adversarial clips for text-to-image synthesis." CVPR 2023

[2] "Towards universal fake image detectors that generalize across generative models." CVPR 2023

---

### Comment · Area_Chair_DsWR · 2024-08-11

Dear Reviewers,
Authors have carefully prepared their rebuttal addressing the concerns you have raised. Please check the rebuttals and join the discussion about the paper.
Thanks,
Your AC

---

### Decision · Program_Chairs · 2024-09-25

**Decision:**

Accept (poster)

**Comment:**

All reviewers found this paper very novel and interesting.  The authors presented a strong response to the comments raised by reviewers.